# Integrative taxonomy of *Haemaphysalis* (Acari: Ixodidae) from the Western Ghats, India: Morphological and molecular characterization and implications

**K.R. Reshma**[1]*, **K. Prakasan**[1], **R. Aswathi**[1], **A.P. Ranjith**[2,3,4]

**1** Department of Zoology, Maharaja's College, Mahatma Gandhi University, Ernakulam, Kerala, India, **2** Integrative Insect Ecology Research Unit, Department of Biology, Faculty of Science, Chulalongkorn University, Bangkok, Thailand, **3** Kālinga Foundation, Agumbe, Karnataka, India, **4** Centre for Biodiversity Genomics, University of Guelph, Ontario, Canada

* reshma.kalarikkal.reghu@gmail.com

## Abstract

*Haemaphysalis* ticks (Ixodidae) are important vectors of pathogens affecting humans and animals, yet their genetic diversity and subgeneric relationships remain insufficiently resolved. We combined detailed morphological examination with cytochrome c oxidase subunit I (COI) -based molecular analyses to provide an integrative taxonomic assessment of the genus. A total of 239 ticks representing ten species were collected from three zones of the Western Ghats, Kerala, India, with *Haemaphysalis* (*Kaiseriana*) *spinigera* Neumann, 1897 and *Haemaphysalis* (*Kaiseriana*) *turturis* Nuttall & Warburton, 1915 being the most abundant. Morphological identifications were confirmed by COI sequence analyses, generating 29 new sequences from 10 species, including five sequenced for the first time. These data were analysed together with publicly available sequences to infer phylogenetic relationships. Maximum Likelihood and Bayesian analyses produced largely congruent topologies but did not support the monophyly of most subgenera. Instead, the subgenus *Allophysalis* was recovered as sister to the remaining *Haemaphysalis*, with *H.* (*Allophysalis*) *kopetdaghica* Kerbabaev, 1962 forming an exception that may represent a distinct lineage requiring further taxonomic evaluation. Species delimitation analyses based on ASAP and PTP recovered 55 and 59 MOTUs, respectively, showing high congruence between methods. Inter-subgeneric divergence ranged from 12–19%, where most species exhibited less than 2% intraspecific variation, several taxa showed patterns consistent with cryptic diversity. Population genetic analyses of *H.* (*Kaiseriana*) *bispinosa* Neumann, 1897 revealed high within-population variation, moderate among-population differentiation, and 20 geographically structured haplotypes across 10 countries. Overall, our results support the utility of COI for *Haemaphysalis* identification at the species level and highlight the need for broader integrative approaches to resolve persistent taxonomic uncertainties in this medically important genus.

**Data availability statement:** All newly generated sequences for this study are deposited in the NCBI GenBank repository (https://www.ncbi.nlm.nih.gov/) with the following accession numbers: PQ410419, PQ410420, PQ410421, PQ410422, PQ410423, PQ410424, PQ410425, PQ432748, PQ432749, PQ432788, PQ432789, PQ432790, PQ432791, PQ436027, PQ436028, PQ436029, PQ43652, PQ436522, PQ444036, PQ444037, PQ444038, PQ444039, PQ444040, PQ451931, PQ451932, PQ451933, PQ451934, PQ585789 and PQ585790.

**Funding:** This study was financially supported by the Council of Scientific and Industrial Research, India (https://www.csir.res.in) in the form of a Senior Research Fellowship award (08/724(0004)2019-EMR-I) received by KR for executing the field survey, data collection and analyses. No additional external funding was received for this study. The funder had no additional role in study design, decision to publish, or preparation of the manuscript.

**Competing interests:** The authors have no relevant financial or non-financial interests to disclose. A.P. Ranjith is an academic editor of PLOS One, but took no part in the peer review and decision-making process for this paper.

## Introduction

*Haemaphysalis* Koch, 1844 (Acari: Ixodidae) is the second largest metastriate tick genus, comprising approximately 176 species worldwide and occurring on all continents except Antarctica [1–4]. Species of this genus are widely distributed across India and parasitize a broad range of terrestrial vertebrates, making *Haemaphysalis* one of the most species-rich ixodid lineages in the country [5]. Numerous studies published between 1997 and 2024 have documented the occurrence of *Haemaphysalis* ticks from different regions of India [6]. Of the 125 tick species reported from India, 52 belong to *Haemaphysalis*, with 16 species restricted to southern peninsular India [5,7]. Based on morphological characteristics, Hoogstraal and Kim [8], classified *Haemaphysalis* into three structural groups: structurally primitive, structurally intermediate, and structurally advanced. Among these nine, namely *Aboimisalis*, *Aborphysalis*, *Allophysalis*, *Haemaphysalis*, *Herpetobia*, *Kaiseriana*, *Ornithophysalis*, *Rhipistoma*, and *Segalia* are distributed across India [9].

Surveillance studies across the country consistently report multiple congeners, including *H.* (*Aborphysalis*) *kyasanurensis* Trapido, Hoogstraal & Rajagopalan, 1964, *H.* (*Kaiseriana*) *aculeata* Lavarra, 1904, *H.* (*Kaiseriana*) *bispinosa* Neumann, 1897, *H.* (*Kaiseriana*) *cuspidata* Warburton, 1910, *H.* (*Kaiseriana*) *intermedia* Warburton & Nuttall, 1909, *H.* (*Kaiseriana*) *spinigera* Neumann, 1897, *H.* (*Kaiseriana*) *shimoga* Hoogstraal & Trapido, 1964, *H.* (*Kaiseriana*) *turturis* Nuttall & Warburton, 1915, and *H.* (*Ornithophysalis*) *minuta* Kohls, 1950 especially in human-livestock-wildlife interface regions of the Western Ghats [5,10,11]. Among these, *H.* (*Kaiseriana*) *spinigera* and *H.* (*Kaiseriana*) *turturis* serves as the principal vectors of Kyasanur Forest Disease Virus (KFDV), a high-consequence zoonotic pathogen endemic to this region [11]. Several other species including *H.* (*Aborphysalis*) *kyasanurensis*, *H.* (*Kaiseriana*) *aculeata*, *H.* (*Kaiseriana*) *bispinosa*, *H.* (*Kaiseriana*) *cuspidata*, *H.* (*Kaiseriana*) *intermedia*, *H.* (*Kaiseriana*) *kinneari* Warburton, 1913, *H.* (*Kaiseriana*) *wellingtoni* Nuttall & Warburton, 1908 and *H.* (*Ornithophysalis*) *minuta*, have also been reported to harbour KFDV and frequently co-occur in KFD affected areas, contributing to the enzootic maintenance of tick-borne infections [12–15]. In addition to KFDV, *Haemaphysalis* ticks are associated with other viral pathogens, including Kaisodi virus, Ganjam virus, and Bhanja virus [7]. They are also known to harbour spotted fever group *Rickettsia* species such as *Rickettsia raoultii*, *Rickettsia hoogstralii;* bacterial and protozoan pathogens including *Anaplasma bovis* Donatien & Lestoquard, 1936, *Theileria orientalis* (Yakimoff & Soudatchenkoff, 1931), and *Babesia* spp. [16–21].

Despite their epidemiological importance, the taxonomy of *Haemaphysalis* in India remains unresolved in several respects. The high species diversity of the genus and its confirmed role in pathogen transmission underscore the need for a precise and unambiguous taxonomy framework, particularly in regions where human, livestock, and wildlife habitats overlap [5,7,10,11]. However, taxonomic studies of Indian *Haemaphysalis* ticks have largely relied on classical morphological approaches [22]. Reliable species-level identification using morphology alone is often difficult due to subtle overlapping geographic distributions, and the presence of cryptic species [22]. To overcome these limitations, integrative approaches combining morphology

with mitochondrial and nuclear markers, such as COI, 16S rRNA, and 18S rRNA, have increasingly been applied [23]. These approaches have refined tick systematics in India by revealing distinct genetic lineages and pseudocryptic species complexes within several ixodid genera [22]. In *Haemaphysalis*, COI-based DNA barcoding has proven particularly effective for species-level identification and for resolving species boundaries [24]. Integrative taxonomy therefore plays a critical role in accurately documenting *Haemaphysalis* diversity, assigning pathogen associations to specific tick species, and informing targeted surveillance and control strategies [25]. Such approaches are especially relevant for vector borne disease risk mapping, prioritization of vaccination and acaricide interventions, and early detection of emerging or invasive tick lineages in the Western Ghats and other endemic regions.

In this study, we combine morphological identification with COI-based molecular characterization and phylogenetic analyses to refine species delimitation and clarify evolutionary relationships within *Haemaphysalis*. Specifically, we aim to improve species-level classification, evaluate subgeneric relationships using inter- and intraspecific sequence variation, and assess haplotype diversity and population genetic structure of *H.* (*Kaiseriana*) *bispinosa* across 10 countries. This work represents the first comprehensive COI-based molecular phylogenetic assessment of *Haemaphysalis* and provides an integrative framework for addressing persistent taxonomic and epidemiological questions in this medically important genus.

## Materials and methods

### Sample collection and morphological identification

Tick specimens were collected from three geographically separated zones (~180 km apart): Kottiyoor, Sholayar and Ponmudi representing the northern, central, and southern regions of the Southern Western Ghats in Kerala, India (Fig 1). The permit for collection of specimens in the mentioned field sites was approved by The Principal Chief Conservator of Forests (Wildlife) and Chief Wildlife Warden, Government of Kerala, India, vide order no. KFDHQ-1653/2020-CWW/WL 10. Sampling was conducted between January and December 2024 by flag dragging on forest floor and vegetations, and did not include any human participants, vertebrate animals, or experimental manipulation of wildlife; therefore, ethical approval and informed consent were not required. Geographic coordinates of each sampling sites were recorded using a handheld Garmin eTrex 32 global positioning system. Collection localities were mapped using QGIS version 3.30.3 ([http://www.qgis.org](http://www.qgis.org)) (Fig 1).

Collected ticks were immediately preserved in 70% ethanol and stored at −20°C until further analysis. Morphological examination was performed using a Leica S8 APO stereomicroscope. Species identification followed the dichotomous keys of Trapido et al. [26] and the species description provided by Geevarghese & Mishra [7]. Taxonomic validity was verified using Robbins et al. [4] and morphological terminology followed Geevarghese & Mishra [7].

### Light and scanning electron microscopy

Representative adult and nymphal specimens were photographed using a Leica DMC2900 digital camera mounted on a Leica M205A stereomicroscope with Leica Application Suite software (LAS, Version 4.8). Detailed examination of nymphal external morphology, including, palps, basis capitula, and coxal segments, was performed using scanning electron microscopy (SEM). Selected specimens were processed following standard protocols [27] and examined using a Zeiss Evo 18 SEM. Micrographs were captured using SmartSEM software and processed in Adobe Photoshop CS8 to remove stacking artefacts and standardize background appearance.

### Molecular procedures

For molecular confirmation, representative specimens from each sampling zone were selected. Genomic DNA was extracted using the DNeasy Blood and Tissue Kit (Qiagen), following the manufacturer's spin-column protocol for animal

 

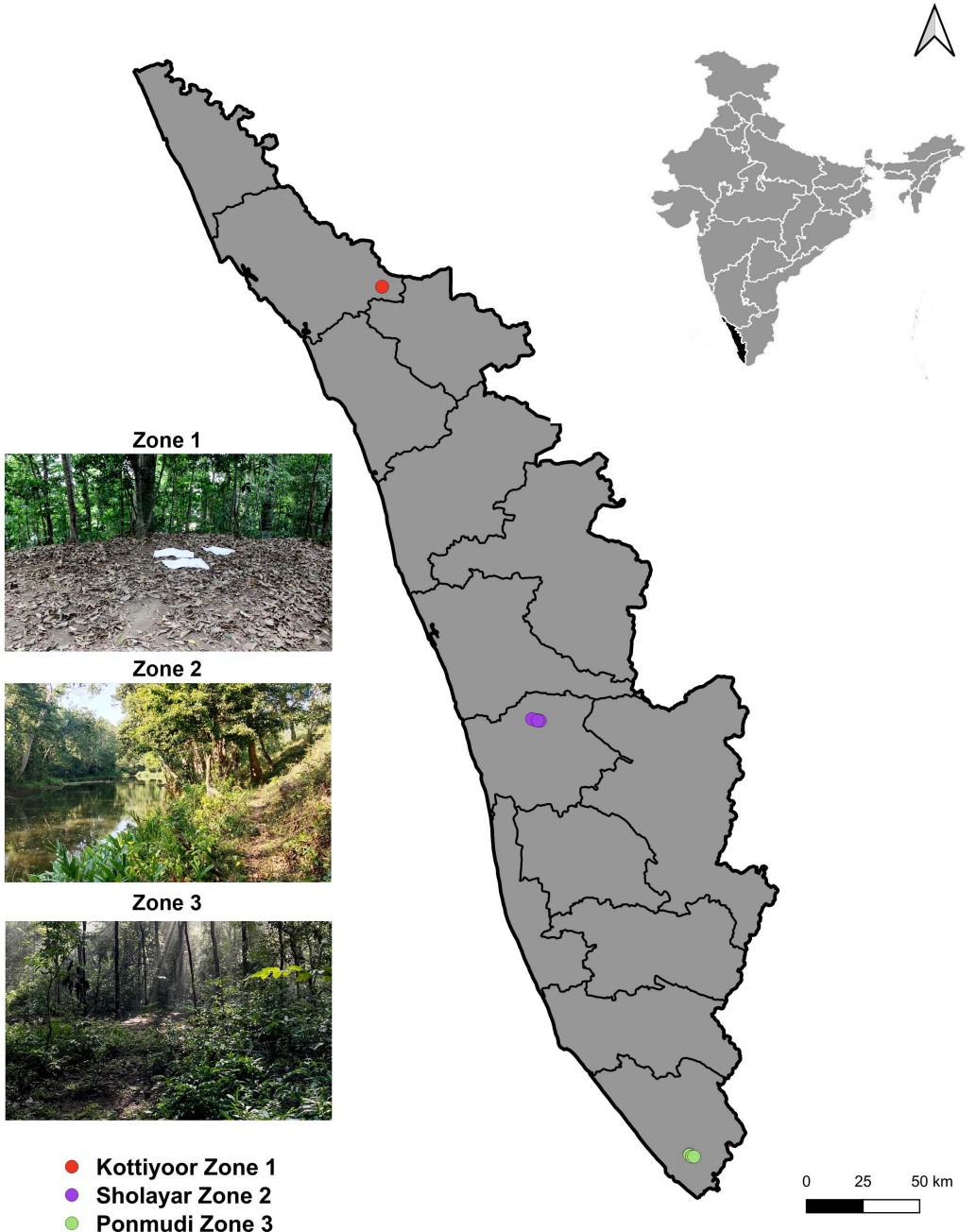

**Fig 1. Map showing the three sampling zones in Kerala, India.** Kottiyoor (Zone 1), Sholayar (Zone 2), and Ponmudi (Zone 3). Publicly available base maps were obtained from the Natural Earth database (https://www.naturalearthdata.com/) and analysed using QGIS version 3.22.12.

tissues. Partial COI gene fragments were amplified using the primer pair LCO1490 and HCO2198 [28] with Sigma Aldrich ReadyMix™ Taq polymerase containing $MgCl_2$.

PCR amplification consisted of an initial denaturation at 94°C for 2 min, followed by 35 cycles of denaturation at 94°C for 10 s, annealing at 45–53°C for 30 s, and elongation at 72°C for 45 s, with a final extension at 72°C for 5 min. PCR products were visualized on 2% agarose gels, purified, and sequenced at OmicsGen LifeSciences.

## Taxon sampling

Publicly available COI sequences of *Haemaphysalis* ticks identified to species level were retrieved from GenBank and BOLD. In addition, 29 new COI sequences were generated in this study and deposited in GenBank (S1 Table). The final dataset included all available *Haemaphysalis* COI sequences that passed quality control, along with four species of the sister genus *Alloceraea* (*A. inermis* (Birula, 1895), *A. kitaokai* (Hoogstraal, 1969), *A. kolonini* (Du, Sun, Xu & Shao, 2018) and *A. colasbelcouri* (Santos Dias 1958)), which were used as outgroups based on recent taxonomic revisions proposed by Kelava et al. [2].

## Selection and curation of COI sequences

All publicly available mitochondrial cytochrome c oxidase subunit I (COI) sequences were subjected to a multi-step curation and validation procedure prior to phylogenetic and species-delimitation analyses, due to the known occurrence of misidentifications in public databases.

**Inclusion criteria.** COI sequences were retained only if they: (i) were associated with a peer-reviewed publication or explicit taxonomic reference; (ii) were linked to a vouchered specimen with repository information (iii) included locality data consistent with the known geographic distribution of the species; (iv) were ≥ ~600 bp in length, covering the standard COI barcode region; (v) showed no internal stop codons or frameshifts upon translation; and (vi) were not exact duplicates, in which case a single representative was retained.

**Taxonomic validation.** Species names were cross-checked against recent taxonomic revisions and authoritative identification keys. Sequences lacking voucher information or morphological corroboration were excluded unless they formed a well-supported clade with sequences of confirmed identity.

**Sequence quality control and outlier screening.** All candidate sequences were translated into amino acids to detect potential nuclear mitochondrial pseudogenes (NUMTs) or sequencing artefacts. Reciprocal BLAST searches against curated reference datasets were conducted to confirm taxonomic consistency. Preliminary phylogenetic analyses were used to identify long-branch and topological outliers. Sequences showing anomalous placements inconsistent with morphology, published diagnoses, or known distributions were excluded or reassigned to higher taxonomic ranks. Only sequences that passed all validation steps were retained for downstream analyses.

## Sequence alignment

Newly generated COI sequences were edited, translated and compiled using BioEdit version 7.0.9 [29]. Sequence alignment was performed using the MAFFT web server with FFT-NS-1 algorithm [30]. The final aligned dataset comprised 676 bp and was used for all phylogenetic and distance-based analyses.

## Phylogenetic analysis

Phylogenetic relationships were inferred using both Maximum Likelihood (ML) and Bayesian Information methods based on the COI dataset. ML analyses were conducted in IQ-TREE v2.1.3 [31]. The alignment was partitioned by codon position, and substitution models were selected using PartitionFinder 2.0 [32] under the Bayesian Information Criterion (BIC). The GTR + G substitution model was assigned to the first and third codon position, and HKY + G to the second codon position. Node support was assessed using 1,000 ultrafast bootstrap replicates.

Bayesian inference was performed in MrBayes 3.2.1 [33] with four independent Markov chain Monte Carlo (MCMC) chains runs executed for 50 million generations, sampling every 100 generations. The alignment was partitioned by codon position, but a single substitution model, GTRGAMMA corresponding to the best-fit model identified by ModelFinder, was applied across all partitions. Model parameters were estimated during the analysis. Convergence was assessed by ensuring the average standard deviation of split frequencies (ASDSF) fell below 0.01 and by inspecting effective sample sizes

(ESS > 200) in Tracer [34]. The first 25% of sampled trees were discarded as burn-in. The remaining trees were used to generate a consensus tree and estimate Bayesian posterior probabilities (PP). Consensus trees were visualized in Fig-Tree version 1.4.4 [35] and edited in Adobe Photoshop CS8. Accession numbers and specimen metadata are provided in S1 Table.

### Species delimitation analyses

Species exhibiting unusually high intraspecific or low interspecific genetic distances were further examined using Assemble Species by Automatic Partitioning (ASAP) and single-rate Poisson Tree Processes (PTP). ASAP analyses were conducted using the online server (https://bioinfo.mnhn.fr/abi/public/asap/) under the Kimura 2-parameter (K2P) model. Species partitions were ranked based on ASAP scores, with lower values indicating stronger support. Single-rate PTP analyses were performed using the ML tree in the mPTP web server (https://mptp.h-its.org/#/tree), with default settings (p = 0.001) [36]. Outgroup taxa were excluded from all species delimitation and distance analyses.

### Pairwise genetic distance (p-distance) analysis of species and subgenera of *Haemaphysalis*

Genetic distances were calculated using the Tamura–Nei model in MEGA version 11 [37] and uncorrected p-distances using the dist.dna() function in the R package *ape* [38]. Distance matrix were reshaped into long format using *tidyverse* [39]. For each species, maximum intraspecific and minimum interspecific distances were extracted to assess the presence of a barcoding gap [40]. Barcoding gap plots and boxplots comparing intra- and interspecific distances were generated using *ggplot2* [41], with species labels adjusted using *ggrepel* [42]. Mean pairwise COI p-distances among *Haemaphysalis* subgenera were calculated and converted to percentage values. Reciprocal subgenus comparisons were averaged to ensure matrix symmetry. A color-coded heatmap illustrating inter-subgeneric divergence was generated using *ggplot2* [41] in R.

### Haplotype and population genetic analyses of *Haemaphysalis* (*Kaiseriana*) *bispinosa*

A total of 82 COI sequences of *H*. (*Kaiseriana*) *bispinosa* were retrieved from GenBank and BOLD for haplotype and population genetic analyses (S2 Table). Sequences were filtered to remove low quality or short fragments prior to analysis. The final aligned dataset comprised 567 bp. Genetic diversity indices including the number of haplotypes, haplotype diversity, nucleotide diversity and segregating sites, were calculated using DnaSP version 6. Analysis of molecular variance (AMOVA) with 1,023 permutations and F-statistics for *H*. (*Kaiseriana*) *bispinosa* populations were calculated in Arlequin version 3.5.2.2 using the pairwise distance method, with molecular variance partitioned to assess population differentiation (Fst). Haplotype networks were constructed using the TCS method in PopART version 1.7.

Pairwise genetic distances were calculated using MEGA 11 and converted to percentages. Distance data were processed in R (version 4.5) using the *tidyverse* package [39], excluding self-comparisons and duplicate pairs. Country-level distance distributions were visualized using boxplot generated in *ggplot2* [41] with rotated x-axis labels for clarity.

## Results

### Tick collections and species composition

In total, 239 *Haemaphysalis* ticks were collected from the three sampling zones in the Western Ghats of Kerala, India. Zone 2 contributed the largest proportion of specimens (49.8%), followed by zone 3 (27.6%) and zone 1 (22.6%). Nine taxa were identified to species level, and one additional taxon was identified only to genus level. The identified species were *H*. (*Kaiseriana*) *aculeata*, *H*. (*Kaiseriana*) *bispinosa*, *H*. (*Kaiseriana*) *cuspidata*, *H*. (*Kaiseriana*) *kinneari*, *H*. (*Kaiseriana*) *shimoga*, *H*. (*Kaiseriana*) *spinigera*, *H*. (*Kaiseriana*) *turturis*, *H*. (*Kaiseriana*) *wellingtoni, H*. sp. near *kyasanurensis* and *Haemaphysalis* sp. Nymphs dominated the collections (94.6%), whereas adult stages comprised 5.4%. Collectively,

the most abundant species was *H.* (*Kaiseriana*) *spinigera,* followed by *H.* (*Kaiseriana*) *turturis*; *Haemaphysalis* sp. was least frequent (2.1%) (Fig 2).

## Morphological characterization

*Haemaphysalis* ticks are distinguished from other ixodid genera by their relatively small body size, sub-rectangular capitulum, and typically short, broad palps that give the mouth parts a triangular appearance. The scutum is inornate and lacked eyes. Festoons are present (11), and the anal grooves surround the anus posteriorly. Species-level identification relied on combinations of characters across life stages, including the ventral outline of palpal segments, the form and relative length of the spur on palpal segment 3 (relative to the ventrobasal margin of palpal segment 2), the number and arrangement of infrainternal setae on palpal segment 2, and the presence and morphology of the coxal spurs. In some specimens, additional diagnostic value is provided by dorsal cornua and the shape and length of the dorsal median retroverted spur on palpal segment 3.

**Adults. Males**: Males of *H.* (*Kaiseriana*) *aculeata* show well-developed dorsal cornua approximately equal in length to the dorsal basis capituli (Fig 3C); spatulate spurs on coxa I and trochanter I, and palpal segment 2 approximately 1.5x the length of segment 3 (Fig 3D). Males of *H.* (*Kaiseriana*) *bispinosa* are characterized by a dorsal median elevated spur on

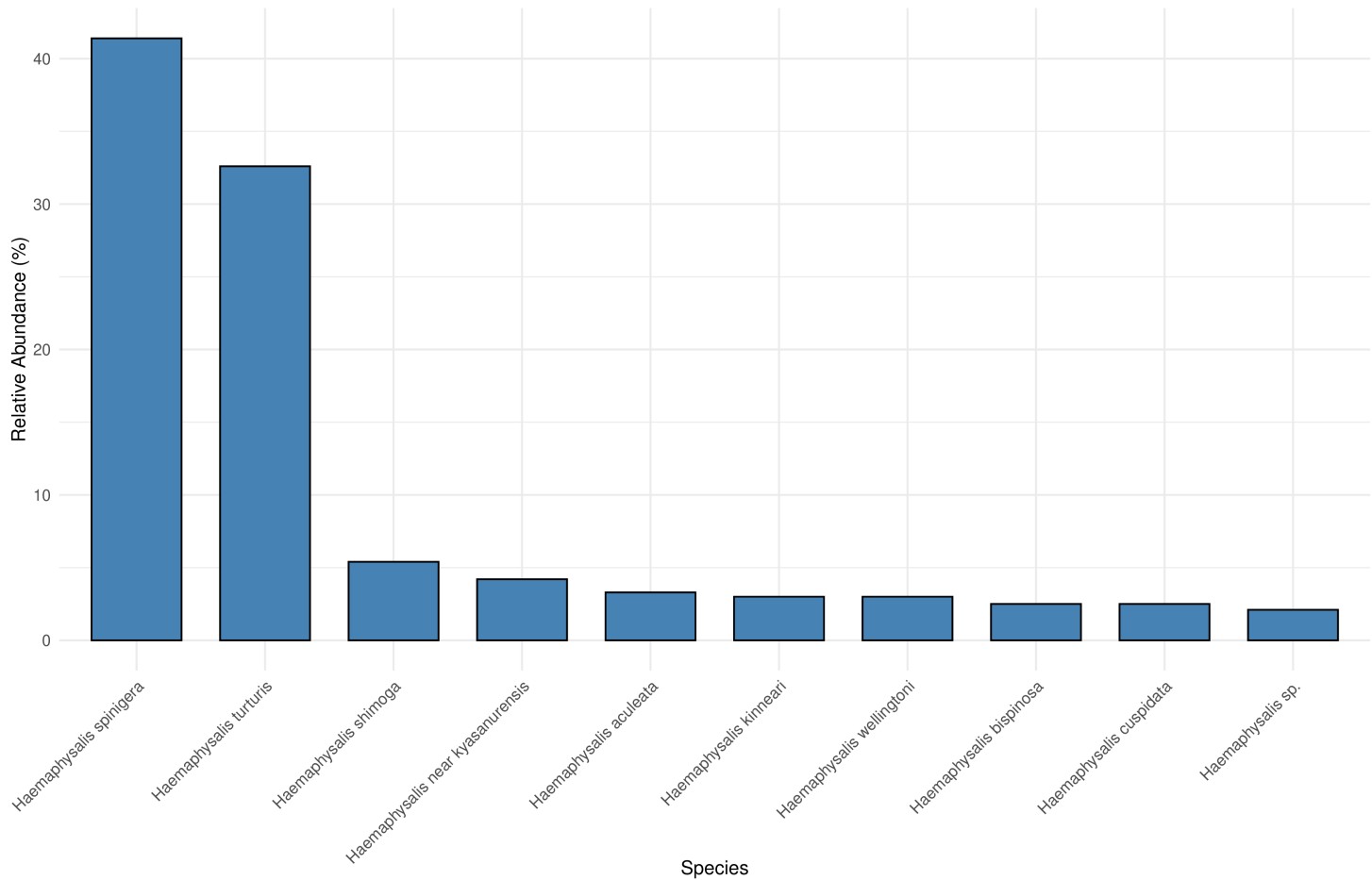

**Fig 2. Relative abundance of *Haemaphysalis* species.** *Haemaphysalis* (*Kaiseriana*) *spinigera* and *H.* (*Kaiseriana*) *turturis* together accounted for >70% of collected specimens.

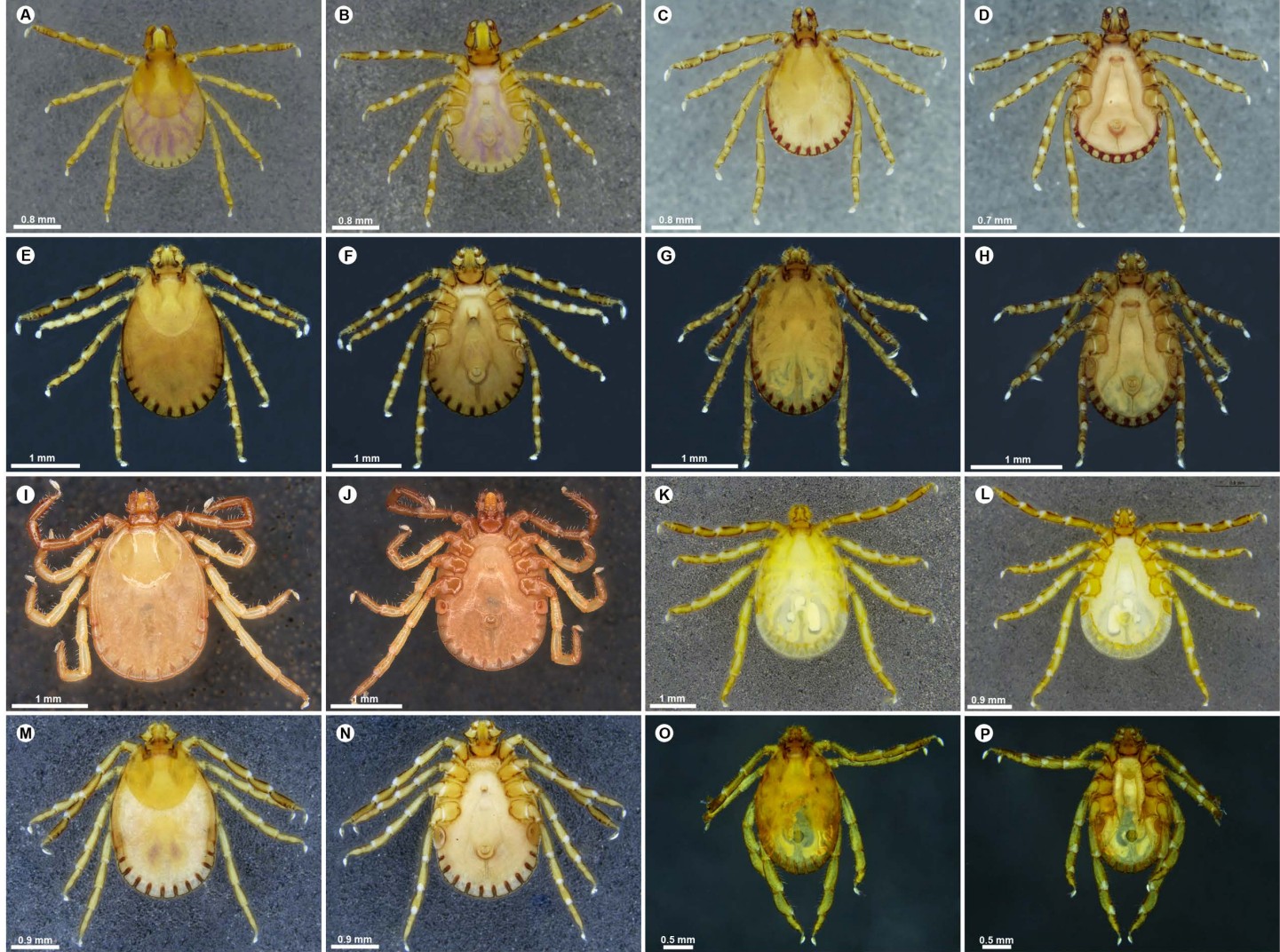

**Fig 3. Dorsal and ventral habitus of adult *Haemaphysalis* species.** (A) *Haemaphysalis* (*Kaiseriana*) *aculeata* female dorsal, (B) *Haemaphysalis* (*Kaiseriana*) *aculeata* female ventral, (C) *Haemaphysalis* (*Kaiseriana*) *aculeata* male dorsal, (D) *Haemaphysalis* (*Kaiseriana*) *aculeata* male ventral, (E) *Haemaphysalis* (*Kaiseriana*) *bispinosa* female dorsal, (F) *Haemaphysalis* (*Kaiseriana*) *bispinosa* female ventral, (G) *Haemaphysalis* (*Kaiseriana*) *bispinosa* male dorsal, (H) *Haemaphysalis* (*Kaiseriana*) *bispinosa* male ventral, (I) *Haemaphysalis* sp. near *kyasanurensis* female dorsal, (J) *Haemaphysalis* sp. near *kyasanurensis* female ventral, (K) *Haemaphysalis* sp. near *kyasanurensis* male dorsal, (L) *Haemaphysalis* sp. near *kyasanurensis* male ventral, (M) *Haemaphysalis* (*Kaiseriana*) *shimoga* female dorsal, (N) *Haemaphysalis* (*Kaiseriana*) *shimoga* female ventral, (O) *Haemaphysalis* (*Kaiseriana*) *shimoga* male dorsal, **(P)** *Haemaphysalis* (*Kaiseriana*) *shimoga* male ventral.

palpal segment 3 and four to five slender, well-spaced infrainternal setae (Fig 3G and 3H). *Haemaphysalis* (*Kaiseriana*) *spinigera* males exhibit a broad salience with a prominent retroverted spur on the ventrobasal-external margin of palpal segment 2, and a markedly elongated, sharp spur on coxa IV exceeding the other coxal spurs (Fig 4C and 4D). Males of *H.* (*Kaiseriana*) *shimoga* share the broad salience and prominent retroverted spur on palpal segment 2 but differ in having a strongly elongated double spur on coxa IV (Fig 3O and 3P). In males of *H.* sp. near *kyasanurensis*, the salience is moderately developed, with a rounded ventrobasal-external margin of palpal segment 2; palpal segment 3 lacks a dorsal basal spur and shows a rudimentary ventral retroverted spur. Coxa I bears a peg-like elongate spur, and coxae II-IV have

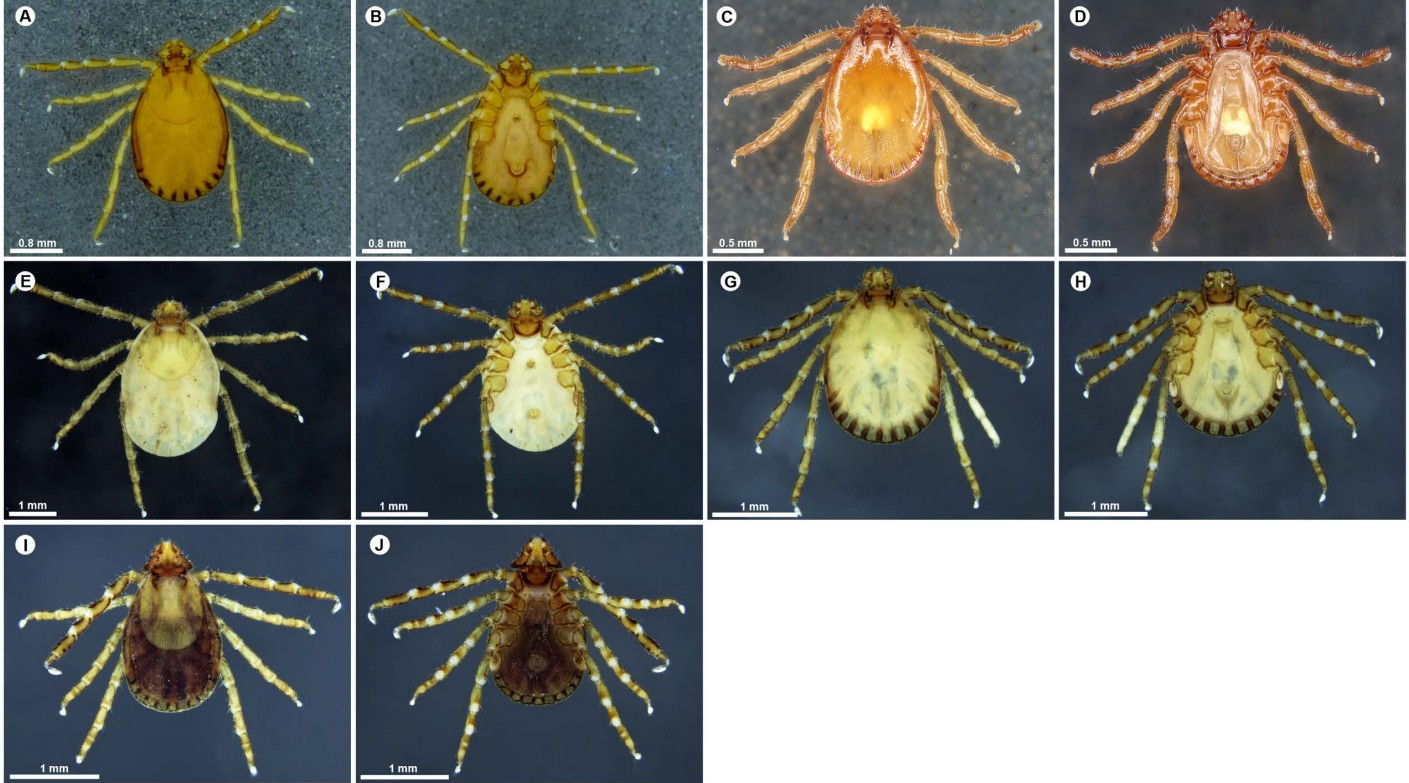

**Fig 4. Dorsal and ventral habitus of adult *Haemaphysalis* species.** (A) *Haemaphysalis* (*Kaiseriana*) *spinigera* female dorsal, (B) *Haemaphysalis* (*Kaiseriana*) *spinigera* female ventral, (C) *Haemaphysalis* (*Kaiseriana*) *spinigera* male dorsal, (D) *Haemaphysalis* (*Kaiseriana*) *spinigera* male ventral, (E) *Haemaphysalis* (*Kaiseriana*) *turturis* female dorsal, (F) *Haemaphysalis* (*Kaiseriana*) *turturis* female ventral, (G) *Haemaphysalis* (*Kaiseriana*) *turturis* male dorsal, (H) *Haemaphysalis* (*Kaiseriana*) *turturis* male ventral, (I) *Haemaphysalis* (*Kaiseriana*) *wellingtoni* female dorsal, (J) *Haemaphysalis* (*Kaiseriana*) *wellingtoni* female ventral.

distinct sharp spurs (Fig 3K and 3L). In *H.* (*Kaiseriana*) *turturis*, palps lack a salience, and palpal segment 3 has a broad, dorsal median, ridge-like projection slightly overlapping the apical margin of palpal segment 2 (Fig 4G and 4H).

**Females**: Females of *H.* (*Kaiseriana*) *aculeata* also show well-developed dorsal cornua (Fig 3A), spatulate spurs on coxa I and trochanter I, and palpal segment 2 approximately twice the length of segment 3 in ventral view (Fig 3B). Females of *H.* (*Kaiseriana*) *bispinosa* share the dorsal median elevated spur on palpal segment 3 and four to five slender, well-spaced infrainternal setae (Fig 3E and 3F). In *H.* (*Kaiseriana*) *spinigera*, females resemble males in salience morphology and the retroverted spur on palpal segment 2, with an additional median retroverted spur on the dorsal margin of palpal segment 3 (Fig 4A and 4B). Females of *H.* (*Kaiseriana*) *shimoga* have a broad salience forming a blunt angle with the ventrobasal and external margins of palpal segment 2; palpal segment 3 possess a prominent ventral retroverted spur and a dorsal median retroverted spur (Fig 3M and 3N). Females of *H.* sp. near *kyasanurensis* have a weaker salience, palpal segment 3 lacks a dorsobasal spur and shows a short ventral spur that only slightly overlaps the apical margin of palpal segment 2. Coxae I-IV each bear distinct sharp spurs (Fig 3I and 3J). In *H.* (*Kaiseriana*) *turturis*, the palps lack a salience, and palpal segment 3 has no dorsobasal-internal spur but displays a broad dorsal projection slightly anterior to the basal margin (Fig 4E and 4F). In *H.* (*Kaiseriana*) *wellingtoni*, the capitulum is broadly deltoid with the juncture of the ventrobasal and external margins of palpal segment 2 forming a rounded angle (Fig 4J); palpal segment 3 has a sharp dorsobasal internal retroverted spur extending approximately half the length of palpal segment 2 (Fig 4I).

**Nymphs.** Nymphs are grouped into two morphological types based on the ventral outline of the palps. Nymphs of *H.* (*Kaiseriana*) *spinigera* (Fig 5O and 5P), *H.* (*Kaiseriana*) *shimoga* (Fig 5M and 5N), *H.* (*Kaiseriana*) *kinneari* (Fig 5I and 5J) and *H.* sp. near *kyasanurensis* (Fig 5K and 5L) typically show broadly flared palps. In contrast nymphs of *H.* (*Kaiseriana*) *turturis* (Fig 6A and 6B), *H.* (*Kaiseriana*) *aculeata* (Fig 5A and 5B), *H.* (*Kaiseriana*) *cuspidata* (Fig 5E and 5F), *H.* (*Kaiseriana*) *bispinosa* (Fig 5C and 5D) and *H.* (*Kaiseriana*) *wellingtoni* (Fig 6C and 6D) lack lateral salience, giving the capitulum a compact appearance. Within the flared-palp group, *H.* (*Kaiseriana*) *spinigera* is distinguished by a retroverted spur on the ventrobasal- external margin of palpal segment 2 (Figs 5P and 8B), whereas in *H.* (*Kaiseriana*) *kinneari*, *H.* sp.

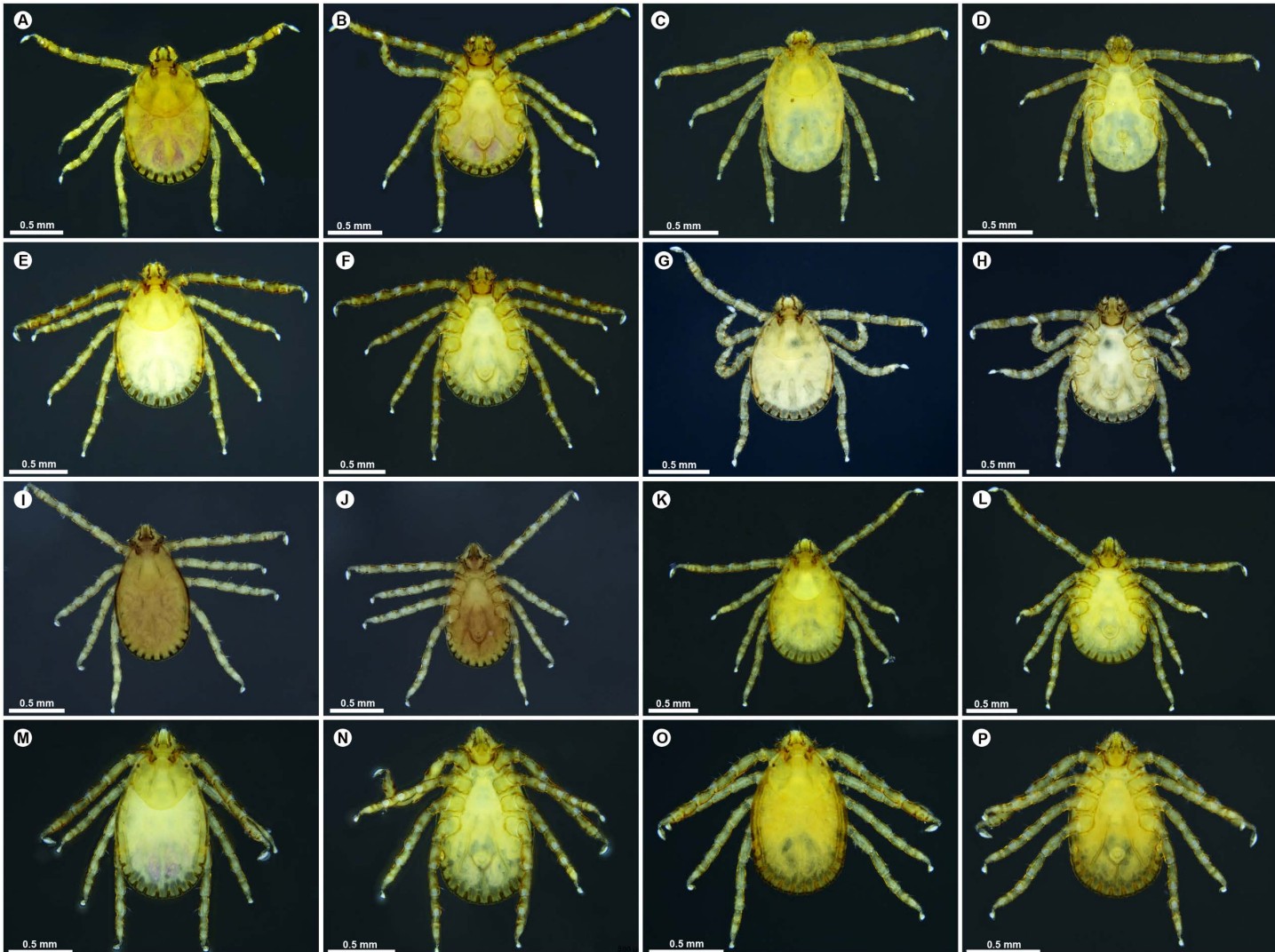

**Fig 5. Dorsal and ventral habitus of nymphal *Haemaphysalis* species.** (A) *Haemaphysalis* (*Kaiseriana*) *aculeata* dorsal, (B) *Haemaphysalis* (*Kaiseriana*) *aculeata* ventral, (C) *Haemaphysalis* (*Kaiseriana*) *bispinosa* dorsal, (D) *Haemaphysalis* (*Kaiseriana*) *bispinosa* ventral, (E) *Haemaphysalis* (*Kaiseriana*) *cuspidata* dorsal, (F) *Haemaphysalis* (*Kaiseriana*) *cuspidata* ventral, (G) *Haemaphysalis* sp. dorsal, (H) *Haemaphysalis* sp. ventral, (I) *Haemaphysalis* (*Kaiseriana*) *kinneari* dorsal, (J) *Haemaphysalis* (*Kaiseriana*) *kinneari* ventral, (K) *Haemaphysalis* sp. near *kyasanurensis* dorsal, (L) *Haemaphysalis* sp. near *kyasanurensis* ventral, (M) *Haemaphysalis* (*Kaiseriana*) *shimoga* dorsal, (N) *Haemaphysalis* (*Kaiseriana*) *shimoga* ventral, (O) *Haemaphysalis* (*Kaiseriana*) *spinigera* dorsal, (P) *Haemaphysalis* (*Kaiseriana*) *spinigera* ventral.

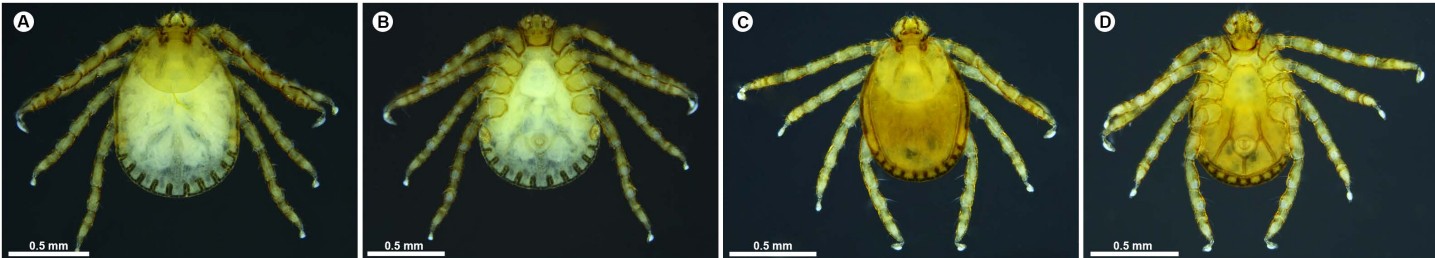

**Fig 6. Dorsal and ventral habitus of nymphal _Haemaphysalis_ species.** (A) _Haemaphysalis_ (_Kaiseriana_) _turturis_ dorsal, (B) _Haemaphysalis_ (_Kaiseriana_) _turturis_ ventral, (C) _Haemaphysalis_ (_Kaiseriana_) _wellingtoni_ dorsal, (D) _Haemaphysalis_ (_Kaiseriana_) _wellingtoni_ ventral.

near _kyasanurensis_, and _H._ (_Kaiseriana_) _shimoga_, the junction of the ventrobasal margin of palpal segment 2 with the externobasal margin form an approximate (blunt to sharp) (Fig 7D-7F).

Among species with a compact capitulum, the ventral external profile of palpal segment 2 is diagnostically informative. _Haemaphysalis_ (_Kaiseriana_) _turturis_ shows pronounced recurvature (Figs 6B and 8C), whereas _H._ (_Kaiseriana_) _bispinosa_ and _H._ (_Kaiseriana_) _wellingtoni_ lack recurvature (Figs 5D, 6D, 7B, 8D). The relative length of ventral spur on palpal segment 3 also aided in identification: approximately three-quarters the length of palpal segment 2 in _H._ (_Kaiseriana_) _cuspidata_ (Figs 5F and 7C)) and _H._ (_Kaiseriana_) _aculeata_ (Figs 5B and 7A); approximately one-half in _H._ (_Kaiseriana_) _turturis_ (Figs 6B and 8C), and approximately one-quarter in _H._ (_Kaiseriana_) _bispinosa_ (Figs 5D and 7B). All examined species have a single dorsal infrainternal seta, whereas the number, thickness, and spacing of ventral infrainternal setae differs among species. Most species have two slender, well-spaced infrainternal setae on palpal segment 2, whereas _H._ (_Kaiseriana_) _wellingtoni_ has four closely-set, feathery infrainternal setae (Figs 6D and 8D), and _H._ sp. near _kyasanurensis_ has a single infrainternal seta (Figs 5L and 7E).

Coxal spurs provide additional diagnostic information among morphologically similar nymphs. In _H._ (_Kaiseriana_) _aculeata_, coxal spurs are strong and decreased progressively in size from coxae I-IV (Fig 5B), whereas in _H._ (_Kaiseriana_) _cuspidata_ coxae II-IV bear ridge-like projections (Fig 5F). In _H._ (_Kaiseriana_) _kinneari_, the spur on coxa I is relatively small and approximately equal in length to (or shorter than) the ventral spur of palpal segment 3 (Fig 5J). Whereas in _H._ (_Kaiseriana_) _shimoga_ it is more prominent and longer than the ventral spur of palpal segment 3 (Fig 5N). Dorsal cornua are generally not diagnostic in nymphs, except in _H._ (_Kaiseriana_) _aculeata_ and _H._ (_Kaiseriana_) _cuspidata_, where cornua length exceeds basal breadth slightly (Fig 5A) and approximately two-fold (Fig 5E), respectively.

Nymphs of _Haemaphysalis_ sp. show a pronounced recurvature of palpal segment 2 (Fig 5G and 5H), and a strong internal ventral retroverted spur on palpal segment 3 extending approximately half way to the ventrobasal margin of palpal segment 2 (Figs 5H and 8A). Additional features include two slender, well-spaced ventral infrainternal setae on palpal segment 2 (Fig 8A), a hypostome dentition of 3/3 (Fig 8A), prominent conical spurs on coxae I-IV, and a conspicuous spur on trochanter I (Fig 5H).

**COI sequences and molecular identification**: The aligned COI dataset included newly generated sequences from nine _Haemaphysalis_ species collected in Kerala. BLASTN results supported the morphological identification of _H._ (_Kaiseriana_) _bispinosa_, _H._ (_Kaiseriana_) _shimoga_, _H._ (_Kaiseriana_) _spinigera_, _H._ (_Kaiseriana_) _turturis, and H._ (_Kaiseriana_) _wellingtoni_, with sequence similarity ranging from 97–100% relative to homologous sequences. In contrast, the sequences of _H._ (_Kaiseriana_) _kinneari_ showed ~97% similarity to _H._ (_Kaiseriana_) _hystricis_ and the sequences of _H._ sp. near _kyasanurensis_ showed >99% similarity to _H._ (_Aborphysalis_) _formosensis_. _Haemaphysalis_ (_Kaiseriana_) _aculeata_ and _H._ (_Kaiseriana_) _cuspidata_ showed 92.48% identity to the sequence of _Haemaphysalis_ sp. and 93.7% identity to _H._ (_Kaiseriana_) _bispinosa_ respectively since, reference COI sequences were not available in the queried databases at the time of analysis. Two sequences (accession no. PQ585789 and PQ585790) could not be confidently assigned to species and were labelled as _Haemaphysalis_ sp.

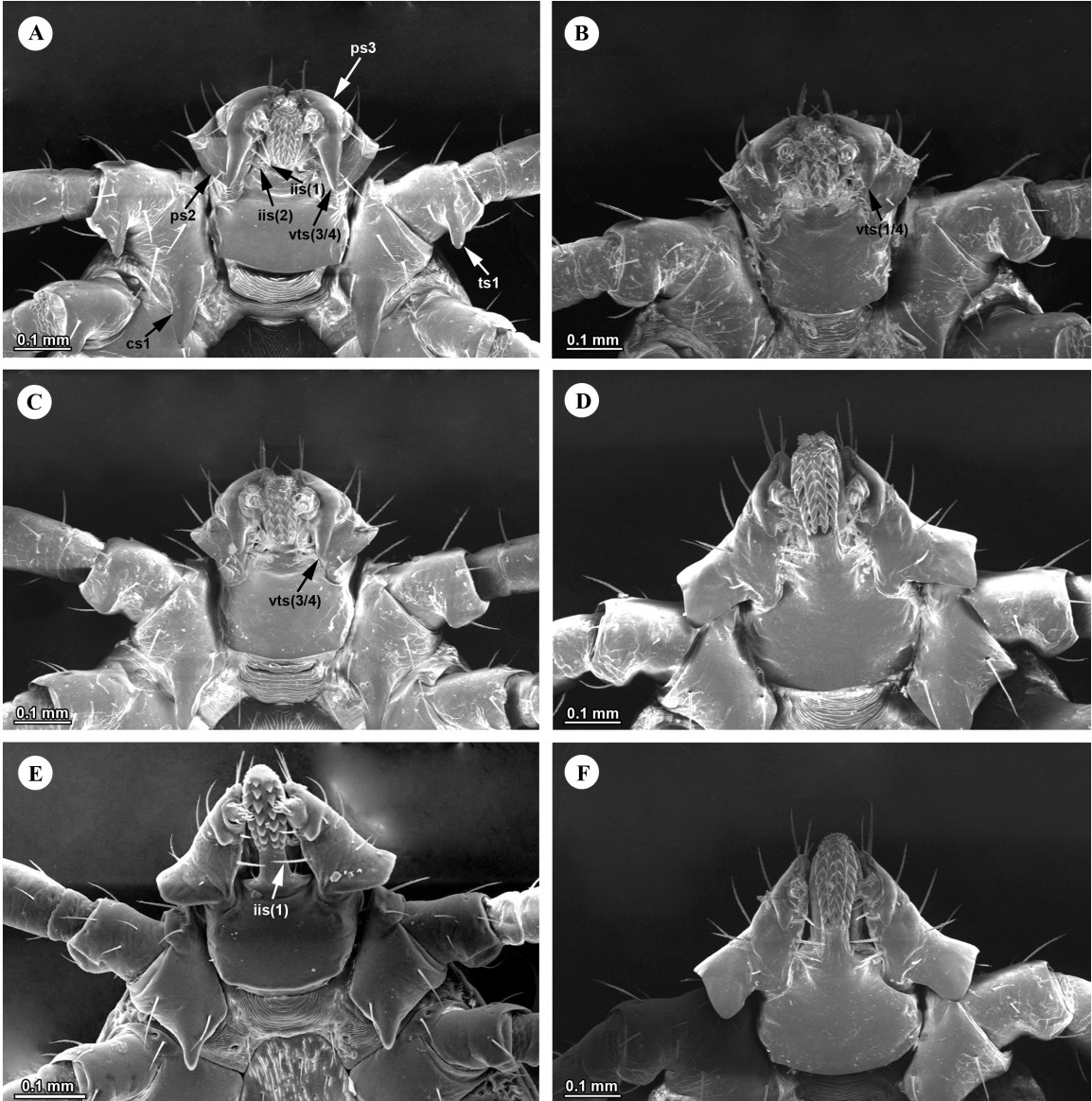

**Fig 7. SEM images of ventral gnathosomal structures in *Haemaphysalis* nymphs.** (A) *Haemaphysalis* (*Kaiseriana*) *aculeata*, (B) *Haemaphysalis* (*Kaiseriana*) *bispinosa*, (C) *Haemaphysalis* (*Kaiseriana*) *cuspidata*, (D) *Haemaphysalis* (*Kaiseriana*) *kinneari*, (E) *Haemaphysalis* sp. near *kyasanurensis*, (F) *Haemaphysalis* (*Kaiseriana*) *shimoga*. iis-infrainternal setae; cs1- coxal spur I; ps2-palpal segment 2; vts-ventral retroverted spur.

## Phylogeny of *Haemaphysalis* and subgeneric relationships

The COI phylogeny comprised 110 sequences, including four *Alloceraea* outgroups. ML and BI analyses recovered *Haemaphysalis* as monophyletic, with *Alloceraea* as the sister (ML = 100%; BI = 1.0) (Figs 9 and S1). Subgeneric classifications were inconsistently monophyletic, with low support (<69%) at several deeper nodes (Fig 9).

A basal *Allophysalis* clade was strongly supported (UF = 100%), but *H.* (*Allophysalis*) *kopetdaghica* grouped with intermediate and advanced subgenera (*Aborphysalis*, *Haemaphysalis*, *Herpetobia*), suggesting a typical placement or

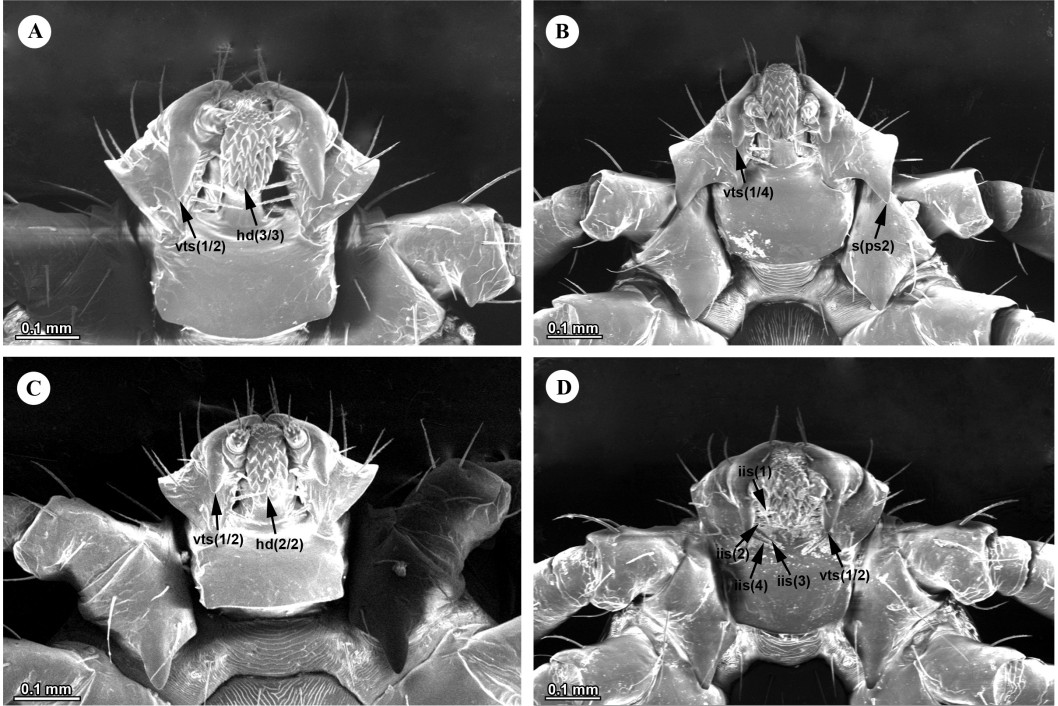

**Fig 8. SEM images of ventral gnathosomal structures in *Haemaphysalis* nymphs.** (A) *Haemaphysalis* sp., (B) *Haemaphysalis* (*Kaiseriana*) *spinigera*, (C) *Haemaphysalis* (*Kaiseriana*) *turturis*, (D) *Haemaphysalis* (*Kaiseriana*) *wellingtoni*. iis-infrainternal setae; hd- hypostomal dentition; ps2-palpal segment 2; vts-ventral retroverted spur.

subgeneric assignment or voucher identity (Figs 9 and S1). *Haemaphysalis* (*Herpetobia*) *nepalensis* clustered with *H.* (*Allophysalis*) *tibetensis*, indicating possible misidentification (Fig 9). The subgenus *Aboimisalis* was monophyletic in ML (UF = 100%) (Fig 9) but not in BI (grouped with *Segalia*) (S1 Fig). The following subgenera *Aborphysalis*, *Gonixodes*, *Haemaphysalis*, *Herpetobia*, *Kaiseriana*, *Ornithophysalis*, *Rhipistoma*, and *Segalia* were polyphyletic (Figs 9 and S1). Within advanced subgenera, *Rhipistoma* formed two clades (bootstrap = 88%), and inclusion of *H.* (*Ornithophysalis*) *minuta* rendered *Ornithophysalis* paraphyletic (Figs 9 and S1). Newly generated sequences clustered within expected species groups with strong support (ML: 90–100%; BI: 1.0) (Figs 9 and S1). *Haemaphysalis* (*Kaiseriana*) *bispinosa* from India, Bangladesh, Indonesia, Malaysia, and Vietnam formed a well-supported clade (ML = 100%) with moderately supported subclades (ML = 46–100%) (Fig 9). *Haemaphysalis* (*Kaiseriana*) *cuspidata* grouped with *H.* (*Kaiseriana*) *bispinosa* (closest relative). *Haemaphysalis* (*Kaiseriana*) *spinigera* clustered with its reference and was sister to *H.* (*Kaiseriana*) *cornigera*, both distant from *H.* (*Kaiseriana*) *shimoga*. *Haemaphysalis* (*Kaiseriana*) *wellingtoni* formed subclades and was sister to *H.* (*Kaiseriana*) *aculeata*, both distant from *H.* (*Kaiseriana*) *turturis* (Fig 9). *Haemaphysalis* sp. near *kyasanurensis* formed a strongly supported clade (ML = 100%) with *H.* (*Aborphysalis*) *formosensis*, sister to *H.* (*Haemaphysalis*) *concinna*. *Haemaphysalis* (*Kaiseriana*) *kinneari* was sister to *H.* (*Kaiseriana*) *hystricis*.

Indian taxa generally formed distinct clades separate from non-Indian sequences (Figs 9 and S1), indicating geographic structuring; East and Southeast Asian taxa (Japan, China, Cambodia, Malaysia) also formed largely non-overlapping clades. BI topology was broadly congruent with ML, differing slightly in the placement of *H.* (*Kaiseriana*) *wellingtoni*, *H.* (*Kaiseriana*) *mageshimaensis*, *H.* (*Kaiseriana*) *hystricis*, and *H.* (*Gonixodes*) *leporipalustris* (Figs 9 and S1).

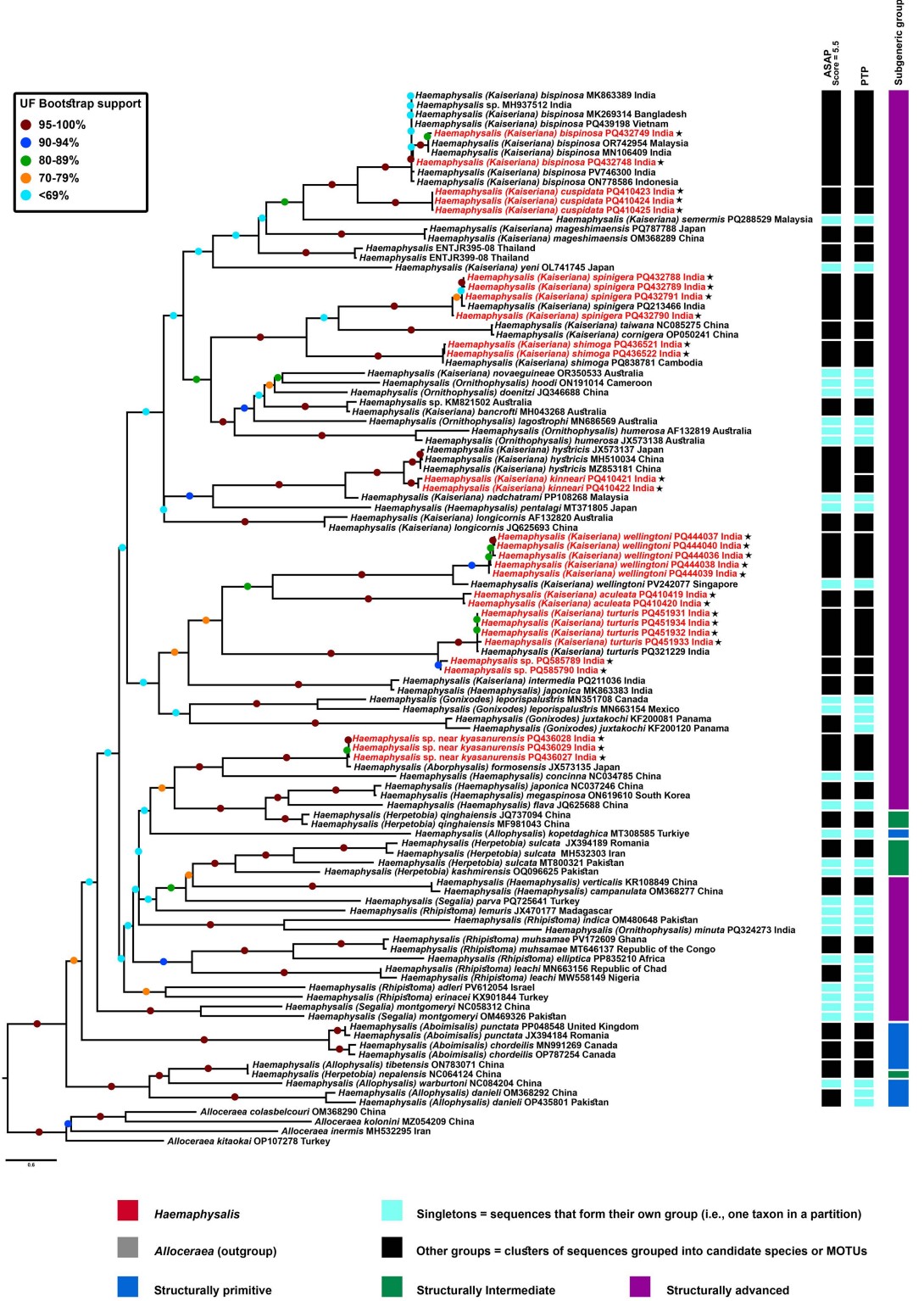

**Fig 9. Maximum Likelihood phylogeny of *Haemaphysalis* species based on COI sequences (676 bp), rooted with *Alloceraea*.** Support values are shown as ultrafast bootstrap (UFBS) percentages. Newly generated sequences are indicated by an asterisk (*). Results of species delimitation (ASAP and PTP) are shown alongside phylogeny.

## Species delimitation analyses (ASAP and PTP)

ASAP and PTP analyses delimited 55 and 59 molecular operational taxonomic units (MOTUs), respectively, with the best ASAP score of 5.5 (Figs 9, S1 and S2). Although *H.* (*Kaiseriana*) *bispinosa* formed multiple subclades in the phylogenetic analyses, ASAP recovered this group as a single species. Several species pairs with low p-distances (<1%) and strong ML support were grouped as single MOTUs, including *H.* (*Kaiseriana*) *taiwana–H.* (*Kaiseriana*) *cornigera*, *H.* (*Haemaphysalis*) *verticalis–H.* (*Haemaphysalis*) *campanulata*, *H.* (*Kaiseriana*) *intermedia–H.* (*Haemaphysalis*) *japonica*, *H.* (*Haemaphysalis*) *japonica–H.* (*Haemaphysalis*) *megaspinosa*, and *H.* (*Allophysalis*) *tibetensis–H.* (*Herpetobia*) *nepalensis* (Figs 9, S1 and S2). In contrast, taxa such as *H.* (*Ornithophysalis*) *humerosa*, *H.* (*Gonixodes*) *leporipalustris*, *H.* (*Haemaphysalis*) *japonica*, *H.* (*Segalia*) *montgomeryi*, and *H.* (*Herpetobia*) *sulcata* showed high divergence (4–15%) and were recovered as separate units, often as singletons (Figs 9, S1 and S2).

Although ML and BI analyses supported distinct clades for *H.* (*Kaiseriana*) *kinneari* and *H.* (*Kaiseriana*) *hystricis*, ASAP grouped them as one unit, whereas PTP delimited them as separate MOTUs (Fig 9). PTP also split individual sequences within *H.* (*Gonixodes*) *juxtakochi*, *H.* (*Rhipistoma*) *leachi* and *H.* (*Allophysalis*) *danieli* into separate units (Fig 9). For taxa that formed subclades within the phylogenies (*H.* (*Kaiseriana*) *bispinosa*, *H.* (*Kaiseriana*) *spinigera*, *H.* (*Kaiseriana*) *shimoga*, *H.* (*Kaiseriana*) *wellingtoni* [excluding the Singapore sequence], *H.* (*Kaiseriana*) *turturis*, *H.* (*Kaiseriana*) *aculeata*) both methods generally supported single species-level units (Figs 9, S1 and S2).

## Pairwise divergence among *Haemaphysalis* taxa

Pairwise COI comparisons showed high similarity (>99%) within most *Haemaphysalis* species (Table 1; S3 Fig), with *H.* (*Kaiseriana*) *bispinosa* exhibiting 98–99% similarity across South and Southeast Asia, indicating genetic homogeneity.

Elevated intraspecific divergence occurred in *H.* (*Ornithophysalis*) *humerosa* (~4%), *H.* (*Herpetobia*) *sulcata* (~2.4% between Romania and Iran; ~10% vs. Pakistan), and *H.* (*Segalia*) *montgomeryi* (~15% between China and Pakistan), suggesting geographic structuring or cryptic diversity (Table 1; S3 Fig). Moderate divergence was observed in *H.* (*Kaiseriana*) *wellingtoni* (~4% between India and Singapore), *H.* (*Gonixodes*) *juxtakochi* (~3%), and *H.* (*Gonixodes*) *leporipalustris* (~8% between Canada and Mexico) (Table 1; S3 Fig).

Several species pairs showed very low interspecific divergence (<1%), including *H.* (*Kaiseriana*) *taiwana–H.* (*Kaiseriana*) *cornigera*, *H.* (*Haemaphysalis*) *verticalis–H.* (*Haemaphysalis*) *campanulata*, *H.* (*Kaiseriana*) *intermedia–H.* (*Haemaphysalis*) *japonica*, and *H.* (*Allophysalis*) *tibetensis–H.* (*Herpetobia*) *nepalensis*, indicating limited genetic differentiation and possible recent divergence or unresolved taxonomy (Table 1; Fig S3).

## Subgenus-level divergence

Mean COI p-distance among subgenera ranged from ~12–19% (Fig 10). Although subgenera were not consistently monophyletic in the phylogenetic analyses, the distance matrix indicated substantial variation among subgeneric comparisons. Most subgenera showed higher divergence from others (15–19%), whereas *Kaiseriana* and *Herpetobia* exhibited comparatively lower mean pairwise distances (~14–16%) with some subgenera, consistent with closer relationships in the COI dataset (Fig 10).

## Barcoding gap and Box-plot analysis

Most taxa with multiple sequences clustered above the diagonal line in the barcoding gap analysis, indicating that minimum interspecific distances exceeded maximum intraspecific distances and supporting species-level separation using COI (Fig 11). *Haemaphysalis* (*Herpetobia*) *sulcata*, *H.* (*Aboimisalis*) *chordeilis*, *H.* (*Kaiseriana*) *hystricis* and *H.* (*Kaiseriana*) *bispinosa* fell near the diagonal line, reflecting narrow barcode gaps consistent with either relatively high intraspecific divergence or low interspecific divergence. In contrast, *H.* (*Haemaphysalis*) *japonica* and *Haemaphysalis* sp. clustered

**Table 1. Pairwise COI sequence divergence among *Haemaphysalis* species.** Categories denote intraspecific comparisons and low interspecific divergence (<1%). Geographic origin indicates sampling locations. Values represent approximate uncorrected pairwise divergence (%); lower values indicate close genetic affinity, whereas higher values reflect moderate to deep divergence.

| Category | Taxon/ Comparison | Geographic origin(s) | COI divergence (%) | Remarks |
|---|---|---|---|---|
| **Intraspecific** | *H. (Kaiseriana) bispinosa* (PQ432748) vs. isolates from Bangladesh, Vietnam, Indonesia, India | South & Southeast Asia | 0.1–0.3 | High similarity (>99%) |
| | *H. (Kaiseriana) bispinosa* (PQ432749) vs. India, Malaysia | South & Southeast Asia | 0.3–2 | High similarity (~98%) |
| | *H. (Ornithophysalis) humerosa* | Australia | 4.1 | Elevated intraspecific divergence |
| | *H. (Herpetobia) sulcata* (Romania vs. Iran) | Europe vs. Middle East | 2.4 | Moderate divergence |
| | *H. (Herpetobia) sulcata* vs. Pakistan | Europe/Middle East vs. South Asia | 11–12 | Deep divergence |
| | *H. (Segalia) montgomeryi* (China vs. Pakistan) | East Asia vs. South Asia | 8.7 | Very high divergence |
| | *H. (Kaiseriana) wellingtoni* (India) | India | 0.45–0.75 | Very low divergence |
| | *H. (Kaiseriana) wellingtoni* (India vs. Singapore) | South Asia vs. Southeast Asia | 3.9–4.3 | Moderate divergence |
| | *H. (Gonixodes) juxtakochi* | Panama | 2.8 | Moderate divergence |
| | *H. (Gonixodes) leporipalustris* (Canada vs. Mexico) | North America | 8.2 | High divergence |
| **Low interspecific (<1%)** | *H. (Kaiseriana) taiwana* vs. *H. (Kaiseriana) cornigera* | China | 0.46 | Very low interspecific divergence |
| | *H. (Haemaphysalis) verticalis* vs. *H. (Haemaphysalis) campanulata* | China | 0.76 | Very low interspecific divergence |
| | *H. (Kaiseriana) intermedia* vs. *H. (Haemaphysalis) japonica* | India | 1.16 | Very low interspecific divergence |
| | *H. (Haemaphysalis) japonica* vs. *H. (Haemaphysalis) megaspinosa* | China vs. South Korea | 0.92 | Very low interspecific divergence |
| | *H. (Allophysalis) tibetensis* vs. *H. (Herpetobia) nepalensis* | China | 0.15 | Very low interspecific divergence |
| | *Haemaphysalis* sp. near *kyasanurensis* vs. *H. (Aborphysalis) formosensis* | Asia | 0–0.6 | Nearly identical sequences |

below the diagonal line, where intraspecific divergence exceeded interspecific divergence, consistent with possible misidentification or unresolved taxonomy. Boxplot analysis further demonstrated that most intraspecific comparisons clustered at low p-distance values (typically <0.05) indicating strong genetic cohesion within species, whereas interspecific distances spanned a broader range (0.05–0.25), highlighting variable degrees of divergence between species pairs (Fig 12).

## Haplotype diversity and population structure of *H. (Kaiseriana) bispinosa*

The haplotype dataset comprised 82 COI sequences (567 bp) of *H. (Kaiseriana) bispinosa* species from 10 countries. The alignment contained 515 conserved sites, 37 polymorphic sites, 25 singleton variable sites, 12 parsimony-informative sites, and 15 gap sites. Haplotype diversity (Hd) was 0.656 +/- 0.054 and nucleotide diversity (π) was 0.008 +/- 0.001. AMOVA indicated that 38.89% of the total genetic variation occurred among populations and 61.11% within populations. The overall fixation index (Fst) was 0.3889 ($p = 0.0000$).

The TCS haplotype network constructed from mitochondrial COI sequences of *H. (Kaiseriana) bispinosa* from 10 countries identified 20 distinct haplotypes forming a single, interconnected network (Fig 13). Two major haplogroups were evident, separated by several mutational steps but remaining connected, indicating intraspecific genetic variation. Hap 1 was the most frequent and centrally positioned haplotype (46 sequences), shared among seven Asian countries. It was

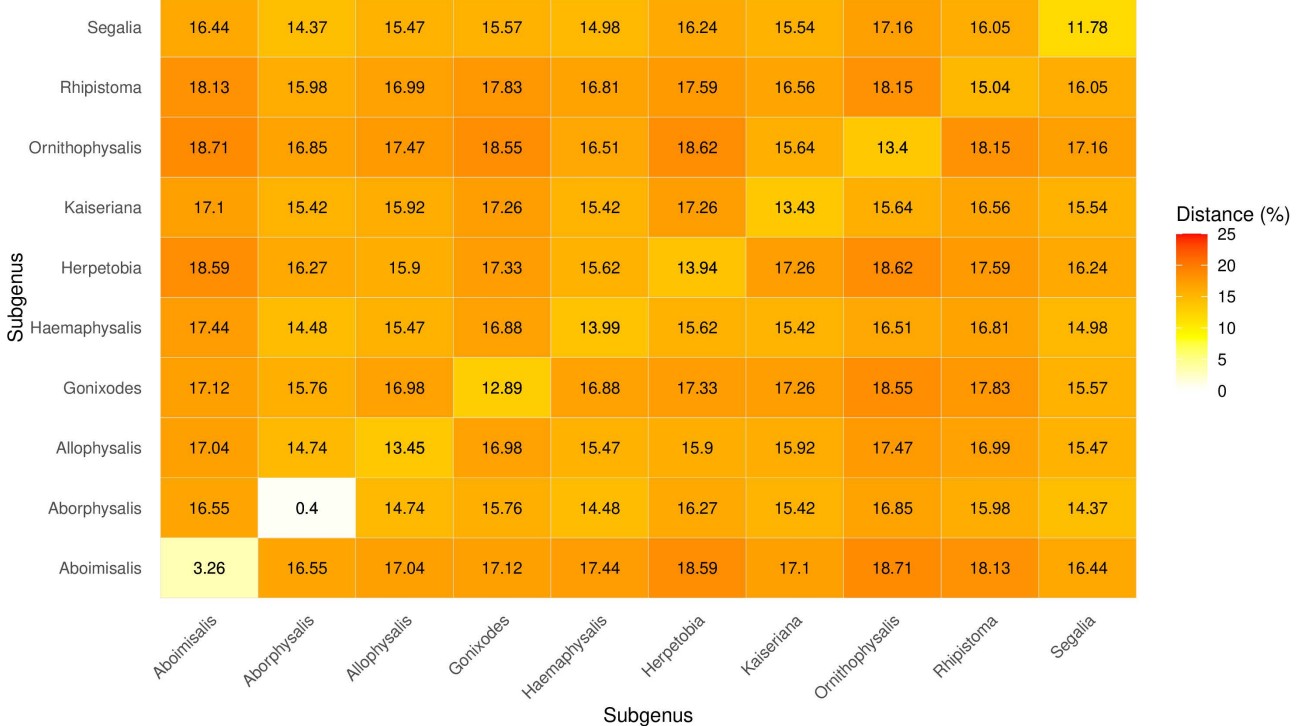

**Fig 10. Heatmap showing mean COI genetic distances (%) among ten *Haemaphysalis* subgenera.**

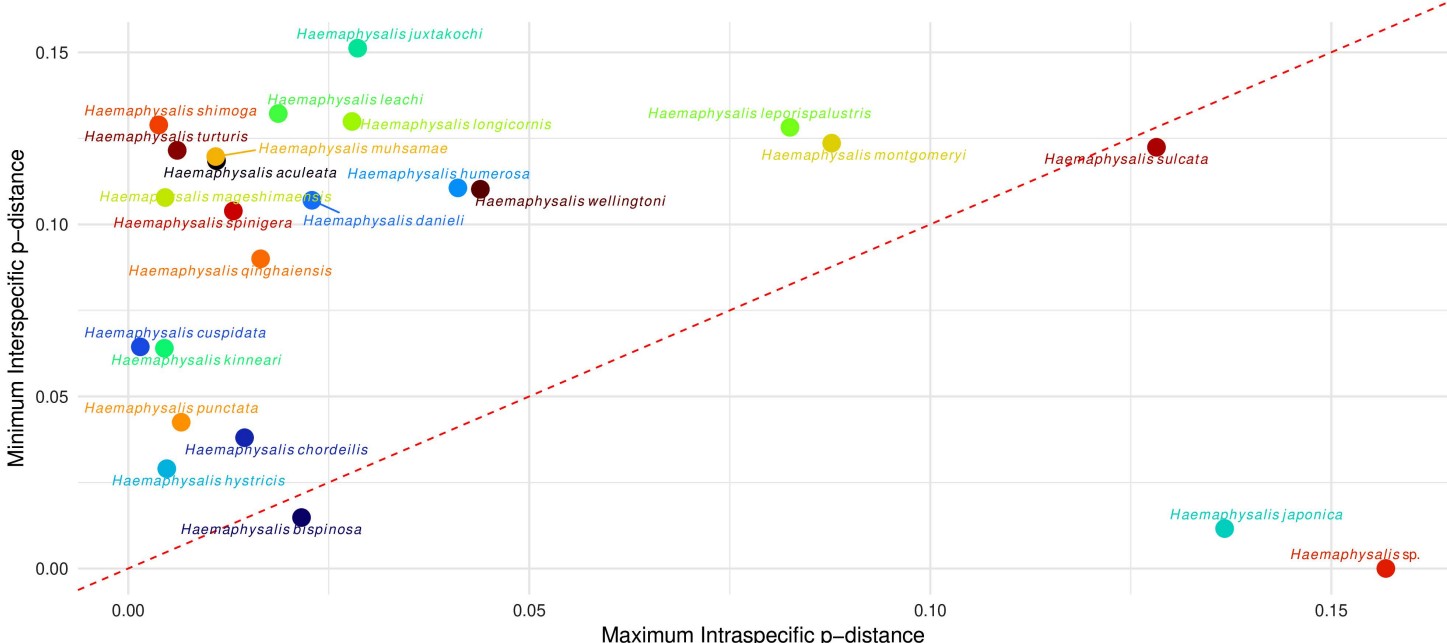

**Fig 11. Barcoding gap plot for *Haemaphysalis* based on COI p-distances.**

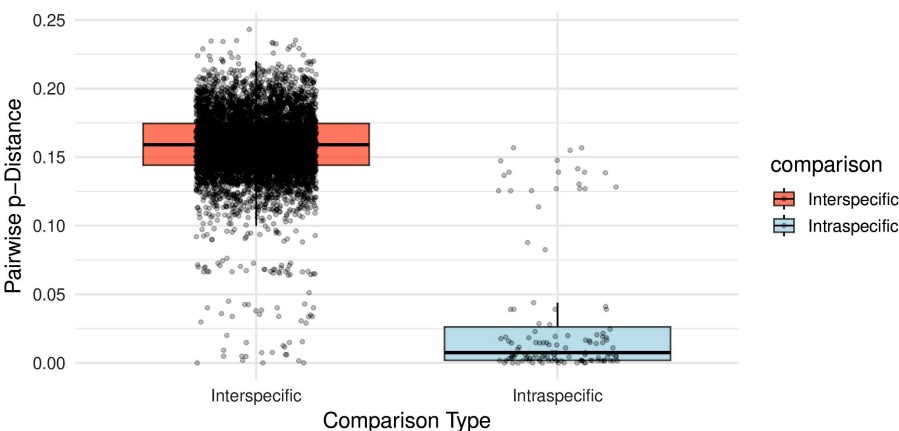

**Fig 12. Boxplot comparison of intraspecific and interspecific COI p-distances in *Haemaphysalis*.**

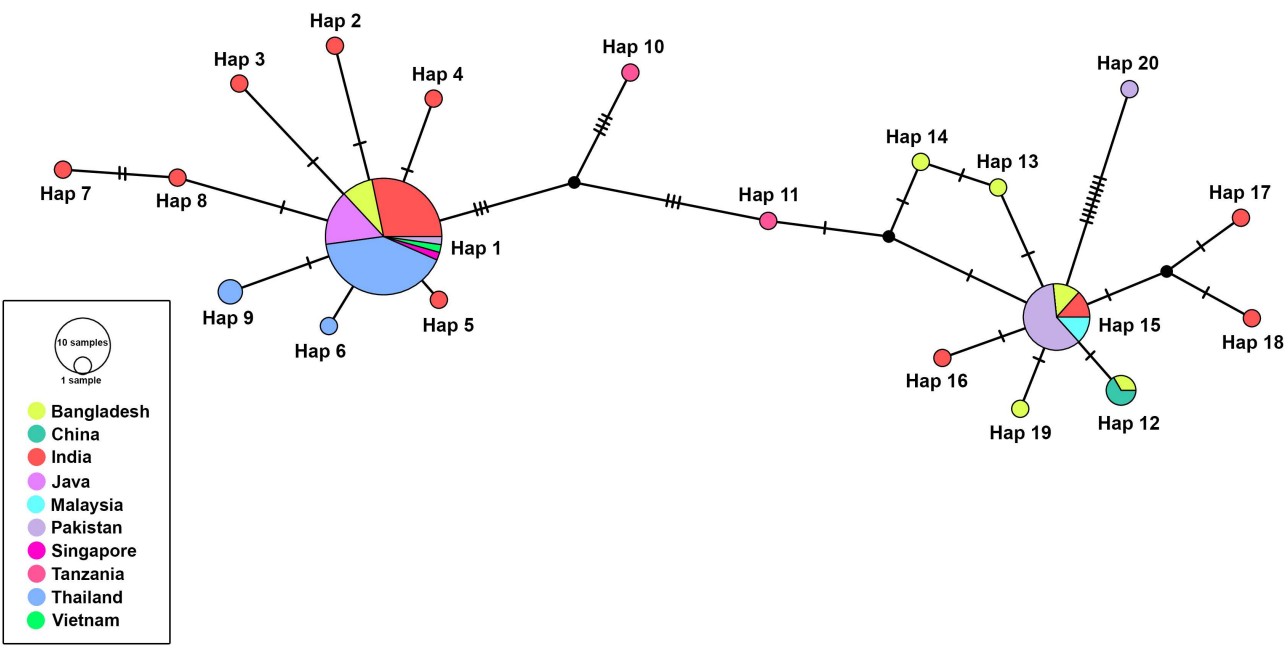

**Fig 13. TCS haplotype network of *Haemaphysalis* (*Kaiseriana*) *bispinosa* COI sequences (567 bp) across 10 countries.**

surrounded by several low-frequency haplotypes differing by one or a few mutational steps, producing a star-like pattern characteristic of recent diversification. A second prominent haplogroup was centered on Hap 15 (15 sequences), which also included sequences from multiple countries and showed several derived haplotypes radiating from it. Peripheral haplotypes represented by one to three sequences, including those from Tanzania, were separated from the central haplo-types by a higher number of mutational steps, suggesting greater genetic divergence relative to Asian populations.

The box plot of pairwise genetic distances among *H.* (*Kaiseriana*) *bispinosa* COI sequences from the 10 countries revealed generally low levels of mitochondrial divergence, with most values below 2% (Fig 14). Median pairwise distances varied among countries, ranging from values close to zero in Java, Singapore, Thailand, and Vietnam to higher medians

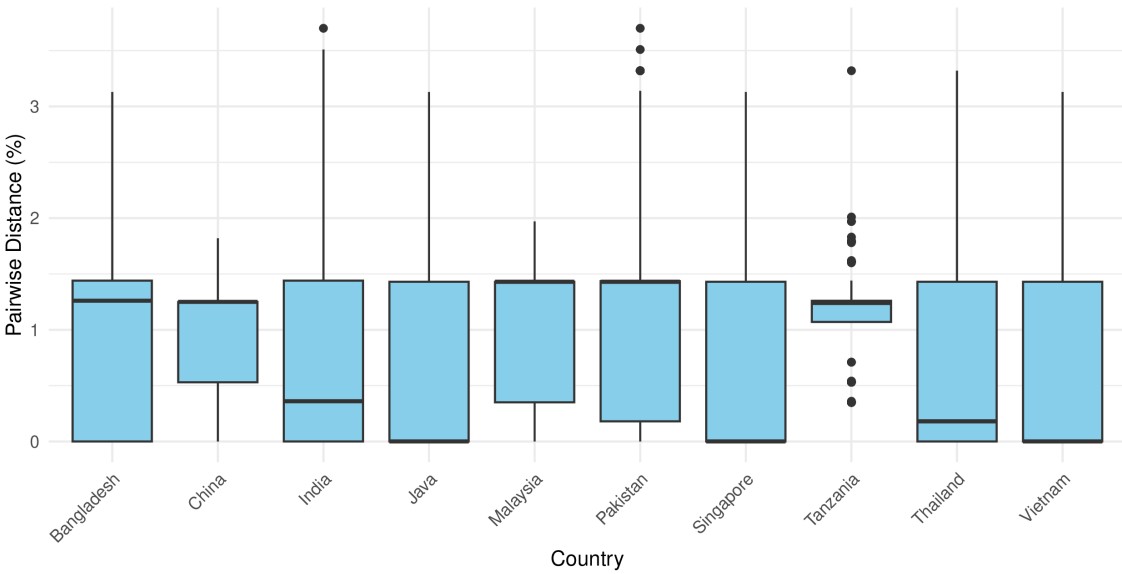

**Fig 14. Boxplot of pairwise genetic distance (%) among *Haemaphysalis* (*Kaiseriana*) *bispinosa* populations from different countries.**

in Bangladesh and Tanzania. India, Pakistan, Bangladesh, and Java exhibited wide interquartile ranges and high maximum values, with outliers exceeding 3%, indicating substantial within-country variability. In contrast, China and Malaysia showed narrower distributions with moderate median distances, reflecting more homogeneous sequence variation. Tanzanian samples displayed a relatively high median pairwise distance coupled with a narrow interquartile range, suggesting consistently elevated divergence within this population.

## Discussion

This study provides an integrative assessment of *Haemaphysalis* ticks with emphasis on species delimitation and the subgeneric placement of lineages inferred from COI sequence data. Among the ticks collected from three zones in the Western Ghats of Kerala, *H.* (*Kaiseriana*) *spinigera* and *H.* (*kaiseriana*) *turturis*, the principal vectors of KFDV, were the most abundant. This dominance is consistent with earlier reports from KFD-endemic areas of southern India, particularly Karnataka and Kerala [7,11,43]. The strong predominance of nymphs over adults likely reflects seasonal stage-specific activity. Previous studies indicate that nymphs increase during the pre-monsoon period, whereas adults emerge around the same time but remain relatively inactive until the onset of monsoon [7,44,45].

Historically, the taxonomy of *Haemaphysalis* has been shaped largely by the work of Hoogstraal and colleagues, who described many species between the 1960s and 1980s [46]. In India, however, the species identifications have relied primarily on morphological keys, especially that of Trapido et al. [26,45]. Several diagnostic characters used in *Haemaphysalis* systematics, including modifications of mouth parts and coxae, have been proposed to reflect host specificity [8]. Variation in these traits across developmental stages and among species may be linked to adaptation to different hosts, producing patterns of parallel evolution between ticks and their hosts [47]. This context helps explain why these structures have remained central to species-level diagnosis within the genus. *Haemaphysalis* has traditionally been divided into 16 subgenera, which Hoogstraal & Kim [8] organized into "structurally primitive," "structurally intermediate," and "structurally advanced" groups. Recent phylogenetic work has questioned the monophyly of these groupings [2,48]. As part of this re-evaluation, *Alloceraea* and *Sharifiella*, previously treated within the structurally primitive assemblage, were elevated to genus rank. *Alloceraea* was recovered as sister to *Haemaphysalis*, whereas *Sharifiella theilerae* was placed outside

*Haemaphysalis* and instead associated with the Rhipicephaline lineage [2,49]; accordingly, *Sharifiella* was not included in our analyses.

Kelava et al. [2] reported that the subgenus *Allophysalis* was paraphyletic. In contrast, our COI based analyses recovered *Allophysalis* as a strongly supported basal lineage sister to the remaining *Haemaphysalis*, although two taxa [*H.* (*Herpetobia*) *nepalensis* and *H.* (*Allophysalis*) *kopetdaghica*] showed discordant placements. The very low pairwise divergence between *H.* (*Herpetobia*) *nepalensis* and *H.* (*Allophysalis*) *tibetensis* (0.15%) suggests possible misidentification of at least one sequence, which differs from the interpretation of Kelava et al. [2]. In addition, *H.* (*Allophysalis*) *kopetdaghica,* reported to exhibit pronounced morphological differences from other members of the subgenus (e.g., high numbers of infrainternal setae: 9 in nymphs and males; 14 in females) [50], clustered with structurally advanced lineages. In our dataset, *H.* (*Allophysalis*) *kopetdaghica* showed 17–20% pairwise divergence from other *Allophysalis* species, comparable to the divergence values reported among subgenera (~19.4%) [2]. This placement warrants targeted re-examination of voucher material and expanded genetic sampling to clarify its taxonomic status and evaluate whether the COI sequence represents an incorrectly identified specimen, an unusually divergent lineage within *Allophysalis,* or a lineage requiring reassignment.

Our results further support the view that the traditional "structural" groupings do not correspond to monophyletic evolutionary units. The subgenus *Herpetobia*, the only member of "structurally intermediate" group, clustered with structurally advanced subgenera inconsistent with the Hoogstraal & Kim framework [8]. Overall, our phylogeny is consistent with earlier evidence for widespread paraphyly or polyphyly among *Haemaphysalis* subgenera and their three structural categories [2,48,51,52].

Across most taxa, pairwise COI p-distances showed a clear separation between intra- and interspecific divergence. Intraspecific distances were generally low, whereas interspecific divergence was substantially higher, and both ASAP and PTP results were broadly consistent with these patterns. However, several morphologically distinct taxa showed unexpectedly low interspecific divergence (<1%). For example, *H.* (*Haemaphysalis*) *japonica* and *H.* (*Haemaphysalis*) *megaspinosa* were treated as a single unit by distance-based and delimitation analyses, despite morphological distinctions. This agrees with previous work reporting low mitochondrial divergence between these taxa (approximately 1% based on mitogenomes) [53]. Similarly, misidentification of female *H.* (*Haemaphysalis*) *campanulata* as *H.* (*Haemaphysalis*) *verticalis* from Jiangxi Province, China [54], underscoring the need for careful re-evaluation of voucher specimens associated with public sequence records. Addressing such issues will improve sequence utility and reduce downstream analytical uncertainty. Specimens identified morphologically as *H.* sp. near *kyasanurensis*; were closely related to *H.* (*Aborphysalis*) *formosensis* from Japan (>99%). *Haemaphysalis* (*Aborphysalis*) *kyasanurensis* described by Trapido et al. [26] is endemic to southwestern India and Sri Lanka and has been considered part of a species complex with *H.* (*Aborphysalis*) *formosensis* Neumann from Formosa and Southeast Asia. These taxa have been separated by multiple morphological characters, including differences in palpal segment proportions, infrainternal setae number (3 versus 4), morphology of ventral spur of palpal segment 3, hypostomal dentition (4/4 versus 6/6), cornua length, scutal punctations, coxal and trochanter spur size and shape, and tarsal proportions [4,26]. Importantly the immature stages of *H.* (*Aborphysalis*) *kyasanurensis* have been described, whereas those of *H.* (*Aborphysalis*) *formosensis* remain undescribed [4,26]. Given the close COI similarity and the likelihood of a species complex, COI alone is insufficient to resolve their taxonomic status. Additional markers and mitogenome-scale data will be required before evaluating synonymy or confirming assignment of the Kerala specimens to *H.* (*Aborphysalis*) *kyasanurensis*.

In contrast to the generally low intraspecific pairwise divergence observed for most taxa, some lineages showed markedly higher within-species divergence (3.5–15%). Among nymphs collected from zone 2, five were morphologically related to *H.* (*Kaiseriana*) *turturis* but formed a distinct lineage with substantial divergence from *H.* (*Kaiseriana*) *turturis* in both the distance matrix and delimitation analyses. This pattern is consistent with the presence of an undescribed sister lineage. Intermediate divergence values (>3%) may indicate cryptic or sibling species in ticks [55,56], and cryptic diversity has

been widely reported in multiple tick genera [22,54,57–66]. In our dataset, *H.* (*Kaiseriana*) *kinneari* and *H.* (*Kaiseriana*) *hystricis* showed >3% divergence; PTP delimited them as distinct units, whereas ASAP grouped them together, suggesting uncertainty that requires further evaluation. Robust resolution of such cases typically requires integrative evidences, including comparative morphology, population genetic structure, ecological and host association data, phylogeography, and analysis of multiple independent genetic markers [22].

Despite these exceptions, the overall pattern in our COI dataset aligns with broader DNA barcoding observations in arthropods, where intraspecific divergence frequently falls below ~2% and interspecific divergence often exceeds ~10% [23,66–69]. Barcoding gap analyses in our study were consistent with this trend and support the use of COI as a practical marker for species-level hypothesis testing in *Haemaphysalis*, while recognizing its limitations in species complexes and recently diverged taxa. This pattern corresponds to the widely recognized DNA barcoding gap, in which genetic distances between species are substantially greater than those within species, thereby supporting the reliability of COI as a marker for species identification. However, barcoding gap thresholds are not universal and may vary among taxonomic groups, being strongly influenced by sampling depth and geographic coverage.

## Population structure of *H.* (*Kaiseriana*) *bispinosa*

The haplotype and population genetic analyses indicated substantial geographic structuring in *H.* (*Kaiseriana*) *bispinosa*. Fst values suggested moderate to strong differentiation among populations across countries (approximately 0.31–0.47), consistent with limited gene flow and geographically structured lineages [70]. Because ticks have limited mobility off-host, dispersal is largely mediated by host movement [71]. *Haemaphysalis* (*Kaiseriana*) *bispinosa* commonly parasitizes domestic ruminants and dogs, and the transport of these hosts may facilitate long-distance spread [72]. These results differ from patterns reported for several other tick genera (e.g., *Amblyomma*, *Dermacentor*, *Ixodes*, *Rhipicephalus*, *H.* (*Kaiseriana*) *longicornis*), where lower structure/genetic variation and greater gene flow have been observed in some studies [70,73].

Network analysis of *H.* (*Kaiseriana*) *bispinosa*, a species traditionally regarded as having an Oriental distribution, revealed shared mitochondrial COI haplotypes among South and Southeast Asian countries, consistent with previous findings by Hornok et al. [74], and additionally included sequences from Afrotropical regions such as Egypt and Tanzania. The star-like configurations around Hap 1 and Hap 15 suggest that these represent ancestral or widely distributed lineages, with surrounding rare haplotypes likely arising through recent mutational events. The presence of shared haplotypes across geographically distant regions indicates substantial gene flow and limited phylogeographic structuring, potentially facilitated by livestock movement and the species' broad host range. The two major haplogroups may reflect historical population subdivision or multiple demographic expansion events. The greater mutational separation of Tanzanian haplotypes supports the influence of continental-scale geographic isolation on mitochondrial divergence, while their integration within the same network confirms their conspecific status. Overall, the haplotype network reveals a pattern of widespread ancestral haplotypes, regional diversification, and historical or ongoing connectivity among *H.* (*Kaiseriana*) *bispinosa* populations across Asia and Africa.

Historically, the geographic range of *H.* (*Kaiseriana*) *bispinosa* was considered to be expanding eastward, with early reports describing it as an introduced species [75]. It is now well established in several Southeast Asian countries, including Malaysia, Indonesia [76], Thailand, Vietnam, and Singapore. However, the full extent of its distribution in Southeast Asia and Africa remains uncertain, partly due to frequent misidentification with *H.* (*Kaiseriana*) *longicornis* [72,77].

The overall low pairwise genetic distances observed across all countries are consistent with intraspecific variation and support the assignment of all analyzed sequences to *H.* (*Kaiseriana*) *bispinosa*. Nevertheless, pronounced regional differences in genetic distance distributions indicate variable levels of mitochondrial diversity and population structuring. High variability and extreme outliers in India and Bangladesh likely reflect broad geographic sampling, historical population subdivision, or the persistence of deeply divergent mitochondrial lineages. In contrast, the relatively narrow distributions

observed in China and Malaysia suggest stronger gene flow or more recent population expansion. Populations from Java, Singapore, Thailand, and Vietnam showed very low median distances but wider overall ranges, consistent with largely homogeneous populations punctuated by occasional divergent haplotypes, potentially associated with founder effects, island biogeography, or human-mediated dispersal via livestock movement. The Tanzanian population exhibited consistently higher divergence compared with Asian populations, supporting the role of continental-scale isolation and restricted interregional gene flow.

The present investigation of mitochondrial COI diversity in *H.* (*Kaiseriana*) *bispinosa* across multiple Asian and African countries provides new insights into the species' genetic structure and geographic distribution. These findings contribute to a better understanding of tick dispersal patterns and may aid in assessing the potential spread of tick-borne pathogens. Although direct evidence for host-mediated dispersal is limited, host movement patterns likely influenced the population structure observed in this study [78]. Long-term surveillance integrating genomic analyses, host association data, and movement tracking will be essential to clarify the mechanisms underlying the spread and gene flow of *H.* (*Kaiseriana*) *bispinosa*.

## Values and limitations

Tick systematics remain largely grounded in morphology, but distinguishing closely related taxa can be challenging when diagnostic traits overlap or vary within species. This problem is well documented in *Haemaphysalis*, where many identifications still rely on subtle morphological differences [53]. COI has high discriminatory power form many taxa and is widely used for species delimitation and identification [54,79,80] and our results support its utility in generating testable species-level hypotheses. However, because our phylogenetic inferences are based on a single mitochondrial locus, the results should be interpreted cautiously. Single-locus analyses can be affected by incomplete taxon and geographic sampling, misidentification in public databases, mitochondrial introgression, endosymbiont-associated selective sweeps, and the amplification of NUMTs [81–84]. For these reasons, we treat the present COI-based framework as preliminary and do not propose any formal taxonomic changes. Future work should incorporate independent nuclear markers (e.g., ITS2, 28S rDNA (D2–D3), 18S, and at least one single-copy nuclear gene such as histone H3 or EF-1α) and, where feasible, genome-scale datasets (e.g., ddRAD-seq, UCEs, or anchored hybrid enrichment) to improve resolution and robustness.

At the same time, molecular tools should complement rather than replace morphology. Because morphological convergence can obscure evolutionary relationships [85–87], integrative approaches combining morphology with multi-locus or genomic data remain the most reliable pathway for resolving systematics, identifying cryptic diversity, and stabilizing classification in medically important tick groups.

## Conclusion

This study integrates morphological diagnosis with COI-based phylogenetics, species delimitation, and population genetic analyses to evaluate diversity and subgeneric relationships in *Haemaphysalis.* Our results support the placement of *Alloceraea* as sister to *Haemaphysalis* and indicate that several traditional subgenera and structural groupings are not monophyletic. Multiple cases of low interspecific divergence, high intraspecific divergence, and potential misidentification in public databases highlight the need for careful voucher-based validation and expanded integrative sampling. The haplotype structure of *H.* (*Kaiseriana*) *bispinosa* indicates geographically structured diversity across countries, consistent with host-mediated dispersal and regional limitation to gene flow. By contributing new COI sequences and an explicitly curated dataset, this work provides a foundation for future taxonomic, ecological, and surveillance studies of *Haemaphysalis* ticks in India and beyond. Further progress will require broader geographic sampling, detailed morphological re-examination of key taxa, and multi-locus or genome-scale data to resolve species complexes and improve confidence in evolutionary and epidemiological interpretations.

## Supporting information

**S1 Table. Specimens used for molecular analysis with their provenances, BOLD information and GenBank accession numbers.** Newly generated sequences are marked with asterisk.
(DOCX)

**S2 Table. *Haemaphysalis* (*Kaiseriana*) *bispinosa* sequences used for haplotype and population study.** Their provenances, BOLD information and GenBank accession numbers. Newly generated sequences are marked with asterisk.
(DOCX)

**S1 Fig. Bayesian inference phylogeny of *Haemaphysalis* ticks based on COI sequences.** Posterior probability values are shown at the nodes to indicate branch support. Sequences are labelled with their GenBank accession numbers and country of origin.
(TIF)

**S2 Fig. ASAP (Assemble Species by Automatic Partitioning) analysis of *Haemaphysalis* based on COI sequences.** The left panel shows partitioning results across different ASAP scores, with each color representing a putative species hypothesis. The right panel depicts the corresponding phylogenetic tree, where yellow and red circles indicate node support values.
(TIF)

**S3 Fig. Heatmap of pairwise COI genetic distances (%) among *Haemaphysalis* species.** The matrix illustrates uncorrected p-distances ranging from 0% (blue, genetically identical or very closely related) to 25% (dark red, highly divergent). Each row and column corresponds to a sequence labelled with species name, GenBank accession number, and country of origin.
(TIF)

## Acknowledgments

The authors thank the Forest Department, Kerala, for permission and arrangements for sample collection. We acknowledge Maharaja's College, Ernakulam, Kerala for infrastructure support; OmicsGen Lifesciences Pvt. Ltd., Ernakulam, Kerala for assistance with DNA sequencing; Division of Arachnology, Sacred Heart College, Thevara, Kerala for microscopy photographs; and Marigenome, Ernakulam, Kerala for SEM imaging. We also thank Dhithya Venkateswaran (Chulalongkorn University, Bangkok) and Anitha Abraham (Maharaja's College, Ernakulam, Kerala) for English language editing and proofreading of the manuscript. We thank the Editor and the anonymous reviewers for their constructive comments and suggestions, which greatly improved the quality and clarity of this manuscript.

## Author contributions

**Conceptualization:** K.R. Reshma, A.P. Ranjith.

**Data curation:** K.R. Reshma.

**Formal analysis:** K.R. Reshma, A.P. Ranjith.

**Funding acquisition:** K.R. Reshma.

**Investigation:** K.R. Reshma.

**Methodology:** K.R. Reshma, A.P. Ranjith.

**Resources:** K.R. Reshma, K. Prakasan, R. Aswathi.

**Software:** K.R. Reshma, A.P. Ranjith.

**Supervision:** K. Prakasan, A.P. Ranjith.

**Validation:** K.R. Reshma, A.P. Ranjith.

**Visualization:** K.R. Reshma, A.P. Ranjith.

**Writing – original draft:** K.R. Reshma, A.P. Ranjith.

**Writing – review & editing:** K.R. Reshma, K. Prakasan, R. Aswathi, A.P. Ranjith.

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
