## [Decision Letter · Decision Letter 0]

27 Oct 2025

PONE-D-25-52984Unravelling the Integrative Taxonomy of Haemaphysalis (Acari: Ixodidae): A Multi-layered Approach with Molecular and Morphological EvidencesPLOS ONE

Dear Dr. Reshma,

Thank you for submitting your manuscript to PLOS ONE. After careful consideration, we feel that it has merit but does not fully meet PLOS ONE’s publication criteria as it currently stands. Therefore, we invite you to submit a revised version of the manuscript that addresses the points raised during the review process.

Please submit your revised manuscript by Dec 11 2025 11:59PM If you will need more time than this to complete your revisions, please reply to this message or contact the journal office at plosone@plos.org. Please include the following items when submitting your revised manuscript:

We look forward to receiving your revised manuscript.

Kind regards,

Maria Stefania Latrofa

Academic Editor

PLOS ONE

Journal Requirements:

4. Please ensure that you refer to Figure 7 in your text as, if accepted, production will need this reference to link the reader to the figure.

5. We note that Figure 1 in your submission contain map images which may be copyrighted. All PLOS content is published under the Creative Commons Attribution License (CC BY 4.0), which means that the manuscript, images, and Supporting Information files will be freely available online, and any third party is permitted to access, download, copy, distribute, and use these materials in any way, even commercially, with proper attribution. For these reasons, we cannot publish previously copyrighted maps or satellite images created using proprietary data, such as Google software (Google Maps, Street View, and Earth). For more information, see our copyright guidelines: http://journals.plos.org/plosone/s/licenses-and-copyright.

Reviewers' comments:

Reviewer's Responses to Questions

**Comments to the Author**

1. Is the manuscript technically sound, and do the data support the conclusions?

Reviewer #1: Partly

Reviewer #2: Partly

2. Has the statistical analysis been performed appropriately and rigorously? 

Reviewer #1: N/A

Reviewer #2: I Don't Know

3. Have the authors made all data underlying the findings in their manuscript fully available?

Reviewer #1: Yes

Reviewer #2: Yes

4. Is the manuscript presented in an intelligible fashion and written in standard English?

Reviewer #1: No

Reviewer #2: No

5. Review Comments to the Author

Reviewer #1: The manuscript addresses a relevant dataset combining COI barcoding, phylogenetic inference, species delimitation methods, and morphological/SEM observations for Haemaphysalis spp. The study has merit and the new sequences and images are valuable contributions, especially considering the epidemiological importance of this tick genus.. However, in its current form, the ms requires substantial improvements in clarity, structure, and balance of interpretations.

Overall, the writing is understandable but there are sections that are too assertive and others that need clearer explanations or more cautious language. The Results and Discussion occasionally overstate the strength of some conclusions, particularly when tree support is low or when different analytical methods do not fully agree. In addition, the description of the methodology would benefit from more detail, especially regarding phylogenetic settings, partition strategies, convergence, and species delimitation parameters.

The morphological component is a strength, but it could be better integrated with the molecular findings . I suggest adding short diagnostic comparisons for key taxa and ensuring that each SEM figure directly supports a statement in the text.

Some species and subgenus names require careful spelling checks,few minor English errors. More neutral wording such as “ suggests”, “is consistent with” or “may indicate” would be more appropriate in several places (e.g. warranting elevation) .

The manuscript also refers to taxonomic implications (such as potential elevation of lineages) in a way that may sound too conclusive for the dataset. I recommend keeping such statements at a hypothesis level, especially when based on a single

marker. The text would benefit from a more explicit acknowledgement of the limitations of COI and of barcoding-gap or MOTU -based interpretations, while still presenting your findings as valuable preliminary evidence.

Minor issues: correcting typographical errors and italics , improving figure captions with clear labels and scale bars, reorganizing some parts of the Discussion to avoid repetition and to focus more clearly on the main message of the study.

I believe the ms has interesting data and could become a solid contribution after a Major Revision. The ms is dense and contains a large amount of information, which at times makes it heavy to read and difficult to follow. Substantial revision is required before it can be considered for publication.

Some specific comments:

L25–27 – suggestion “may represent a distinct lineage that merits further taxonomic evaluation”

L34 – “robust marker” may be misinterpreted ...“useful for species-level hypotheses”

L40–54 – Introduction: too long background

L80–88 – I would suggest a final summarizing sentence: “Here, we combine morphology, SEM, COI phylogeny, and species delimitation to assess diversity .....”

L125: 45-53˚C for 30 seconds (annealing) why such difference? all samples followed the same protocol?

L137: Bayesian information criterion (BIC)

L106: Consider mentioning which structures were prioritized for comparison (e.g., palps, scutum, basis capituli)

L557 discussion: consider reducing repetition of information already stated in introduction, Results. You can shorten the first paragraph by focusing immediately on the key implications of your results

K714: When discussing possible synonymy please add a brief cautionary statement noting

L747: consider adding one sentence acknowledging that barcoding gaps may vary among groups and depend on sampling depth

Reviewer #2: This manuscript analyzes Haemaphysalis ticks using COI barcodes plus morphology, focusing on material collected in Kerala (Western Ghats, India), augmented with public sequences. The authors infer phylogenetic patterns, apply ASAP and PTP delimitation, and analyze haplotype structure in H. (Kaiseriana) bispinosa. While the topic is relevant, the current version has issues of scope, framing, reproducibility, and several technical/formatting problems that must be addressed before the work can be properly evaluated.

Major comments

The title (“Unravelling the Integrative Taxonomy of Haemaphysalis …”) implies a broad, global review of the genus. In reality, new sampling is restricted to Kerala (Western Ghats), and most analyses revolve around a limited set of Indian taxa plus mined public data. The abstract itself centers the work in Kerala/India (e.g., “239 ticks … across three zones of the Western Ghats, Kerala, India”)—this reinforces the mismatch. Please retitle to reflect the actual scope and geographic focus. For example: Integrative taxonomy of Haemaphysalis (Ixodidae) from the Western Ghats, India: COI+ morphology and implications for subgeneric boundaries

Introduction

Introduction needs a focused state-of-the-art for India and public health relevance.

The Introduction expends space on historical taxonomy without synthesizing the current state of knowledge on Indian Haemaphysalis spp., their distribution, and—crucially—their role as vectors (e.g., KFD context). This gap makes the motivation for the study less compelling. Please restructure to:

- Summarize what is known about Haemaphysalis diversity in India, highlighting Western Ghats.

- Clearly articulate medical/veterinary importance in India (which you allude to later), and how better taxonomy impacts surveillance/control.

Overstated claims in the Abstract/Discussion

The abstract asserts broad taxonomic implications (e.g., “H. (Allophysalis) kopetdaghica warranting elevation to a distinct subgenus”), which are not justified by a single mitochondrial locus alone. Such taxonomic acts require integrative evidence (multiple independent loci + morphology across geographic ranges). Please temper wording to “may warrant re-evaluation pending multilocus and morphological revision” and move any taxonomic suggestions to a cautious, hypothesis-level statement.

Material and methods

Please also justify how GenBank records were selected and curated, given the well-known risk of misidentified public sequences. Specify whether each included record was corroborated by a published morphological diagnosis and/or a vouchered specimen (with repository and catalog number), and whether names were cross-checked against recent taxonomic revisions. Clearly state your inclusion/exclusion criteria (e.g., peer-reviewed source; voucher and locality metadata within the species’ known range; full-length COI ≥ ~600 bp; no internal stop codons/frameshifts; ≤X% ambiguous bases; removal of exact duplicates) and the quality-control steps applied (reciprocal BLAST against curated datasets; protein translation to screen for NUMTs; long-branch/topological outlier checks; re-labeling or exclusion of sequences whose placement contradicts morphology or geography).

Please reconcile and fully document the site counts and matrix lengths. In the Methods you state a global COI alignment of 676 bp, whereas the H. bispinosa dataset uses 632 bp; please specify the trimming rules (end clipping, gap-rich columns, handling of ambiguous characters) and indicate which exact alignment file underlies each analysis/figure. For H. bispinosa, the reported counts (253 conserved + 379 polymorphic) do not align with the 14 singleton + 9 parsimony-informative breakdown; under standard settings (e.g., DnaSP with gaps treated as missing), S = singletons + parsimony-informative. Please re-compute and report L, conserved, polymorphic (S), singleton, parsimony-informative, and the number of sites with gaps/ambiguous characters.

Discussion and conclusion

Authors concluded that subgenera are not monophyletic and suggest changes. With COI alone, patterns may reflect limited taxon sampling and other pitfalls.

Please add a concise cautionary paragraph to the Discussion/Conclusions clarifying that your inference of subgeneric non-monophyly is based on a single mitochondrial locus (COI) and should therefore be presented as a working hypothesis pending corroboration. Explicitly acknowledge common pitfalls of COI-only inference—limited taxon and geographic sampling, possible misidentifications in public databases, mitochondrial introgression, endosymbiont-driven selective sweeps, and potential amplification of NUMTs—and adopt a conservative stance that avoids formal taxonomic changes at this stage. Outline what robust corroboration would entail: complementing COI with independent nuclear markers (e.g., ITS2, 28S rDNA D2–D3, 18S) plus at least one single-copy nuclear gene (e.g., histone H3 or EF-1α), and, where feasible, genome-scale datasets (UCEs, ddRAD, anchored hybrid enrichment)

Minor comments

- The abstract mentions “22 geographically structured haplotypes across 11 countries.” However, the Figure 13 legend says “across 12 countries.” Please reconcile (and ensure consistency throughout).

- Line 182: DnaSP (version xx) appears as a placeholder—fill in the exact version used.

- The PCR protocol reports 98 °C denaturation while using “Sigma Aldrich ReadyMix™ Taq”; 98 °C is atypical for standard Taq (common with high-fidelity polymerases). Please verify polymerase type/buffer, and adjust the thermal profile description.

- The UFBoot2 reference line shows an implausible year/volume (“2003; 35: 518–522”)—this looks incorrect for Hoang et al. UFBoot2 (Maybe 2018?). Please fix the bibliographic details to the correct citation.

- Please correct “Bayesian Inference Criterion” to Bayesian Information Criterion (BIC). Also specify how you implemented codon partitioning in IQ-TREE and MrBayes (partition file/commands), given that you report GTR+G for the 1st and 3rd codon positions and HKY+G for the 2nd.

6. PLOS authors have the option to publish the peer review history of their article (what does this mean?). If published, this will include your full peer review and any attached files.

Reviewer #1: No

Reviewer #2: No

---

## [Author Response · Author response to Decision Letter 1]

21 Jan 2026

Dear Editor,

We sincerely thank the editorial office, you and the two reviewers for the thorough and constructive evaluation of our manuscript entitled “Unravelling the Integrative Taxonomy of Haemaphysalis (Acari: Ixodidae)”: A Multi-layered Approach with Molecular and Morphological Evidences. The title has been revised as suggested by the reviewers, as “Integrative taxonomy of Haemaphysalis (Acari: Ixodidae) from the Western Ghats, India: COI+ morphology and implications for subgeneric boundaries”. We appreciate the time and expertise invested in reviewing our work. We have carefully considered all comments and have substantially revised the manuscript to improve clarity, balance of interpretation, methodological transparency, and framing of taxonomic implications. The English language of the manuscript has been carefully checked throughout. We have replaced the section of haplotype diversity of H. bispinosa with a new dataset and the results and discussion section has been revised accordingly. Additionally, we have revised and replaced the following figures, figure 9 (erased two unwanted coloured dots), figure 13 & 14 (according to the revised analysis).

Below, we provide detailed response for the journal requirements posed by editorial office and a point-by-point response to all reviewer comments, indicating how each issue has been addressed in the revised manuscript.

Response to Editorial Office

We thank the Editorial Office for outlining the additional requirements. We have carefully addressed each point to ensure full compliance with PLOS ONE policies, formatting standards, ethics requirements, and copyright guidelines. Our responses are detailed below.

1. Compliance with PLOS ONE style and file-naming requirements

Response:

We have revised the manuscript to fully comply with PLOS ONE formatting and style requirements. The main text, title page, and author affiliations follow the official PLOS ONE templates. File names have been updated according to journal conventions, and all figures, tables, references, and supporting information files were checked for consistency with PLOS ONE guidelines.

2. Code sharing and reproducibility

Response:

We reviewed PLOS ONE’s guidelines on code and software sharing. This study did not involve any author-generated custom code that underpins the findings. All analyses were conducted using established, publicly available software packages (e.g., IQ-TREE, MrBayes, DnaSP, ASAP, PTP), with software versions and analytical parameters fully described in the Methods section. Therefore, there is no proprietary or unpublished code associated with this manuscript. We are happy to provide example command lines or configuration files as Supporting Information if requested to further facilitate reproducibility.

3. Ethics statement in the Methods section

Response:

This study did not involve human participants, vertebrate animals, or any experimental manipulation of wildlife. All specimens were collected from the forest floor as part of routine environmental sampling, without handling, harming, or disturbing any animals. As no humans or animals were involved in the study, and no invasive or regulated procedures were conducted, formal approval from an Institutional Review Board (IRB) or animal ethics committee was not required. Consequently, informed consent was not applicable to this study.

4. Reference to Figure 7 in the text

Response:

We have revised the manuscript to ensure that Figure 7 is explicitly cited in the main text, allowing proper linkage during production.

5. Copyright compliance for map images in Figure 1

Response:

We acknowledge the concern regarding potential copyright restrictions associated with the map in Figure 1. We confirm that Figure 1 was prepared by the authors using shapefiles downloaded from Natural Earth (https://www.naturalearthdata.com/), which provides public-domain geospatial data. Natural Earth datasets are released into the public domain, and their use is fully compatible with PLOS ONE’s Creative Commons Attribution (CC BY 4.0) license. The figure caption has been updated to clearly state that the map was generated using Natural Earth public-domain shapefiles, and no proprietary or copyrighted map sources (e.g., Google Maps, Google Earth) were used. Accordingly, no third-party permissions are required for this figure.

6. Reviewer-recommended citations

Response:

We carefully reviewed all publications suggested by the reviewers. References that were directly relevant and strengthened the scientific background, methodological justification, or interpretation of our results have been incorporated into the revised manuscript. Suggested works that were not directly applicable to the scope of the study were not included, in accordance with journal guidance.

Response to Reviewer 1

We thank Reviewer 1 for the positive assessment of the dataset and for the detailed suggestions that helped us substantially improve clarity, tone, and structure.

General comments

• The manuscript has merit… however, it requires substantial improvements in clarity, structure, and balance of interpretations.

Response:

We agree with this assessment. The manuscript has been thoroughly revised to improve readability, reduce density, and ensure that interpretations are consistently proportional to the strength of evidence. In particular:

• Overly assertive language has been replaced with cautious phrasing (e.g., “suggests,” “may indicate,” “is consistent with”).

Results and Discussion sections were revised to avoid overstating conclusions where node support is low or analytical methods disagree. The Discussion was reorganized to reduce repetition and focus more clearly on the main implications.

• The Results and Discussion occasionally overstate the strength of some conclusions.

Response:

We have carefully reworded these sections to better reflect uncertainty and methodological limitations. Statements implying taxonomic elevation or strong evolutionary conclusions are now explicitly framed as working hypotheses rather than definitive outcomes, particularly when based on COI alone.

• The description of the methodology would benefit from more detail

Response:

We have expanded the Methods section to include:

Explicit phylogenetic settings (models, partitioning schemes, software versions).

Details on convergence diagnostics for Bayesian analyses.

Parameters used in species delimitation analyses (ASAP, PTP), including justification of thresholds. These changes improve reproducibility and transparency.

• The morphological component is a strength, but it could be better integrated

Response:

We have improved integration by:

Adding brief diagnostic morphological comparisons for key taxa and their validation by molecular data in the Results/Discussion. Ensuring that each SEM figure is explicitly referenced in the text and directly supports a stated observation. Clarifying which morphological structures (e.g., palps, scutum, basis capituli) were prioritized for comparison.

• Some species and subgenus names require careful spelling checks

Response:

All taxonomic names were carefully checked for spelling, formatting, and italics. Minor English errors were corrected throughout.

• The manuscript refers to taxonomic implications in a way that may sound too conclusive

Response:

We fully agree. All taxonomic implications are now presented cautiously and explicitly framed as hypotheses requiring multilocus and broader morphological validation. We also added a dedicated paragraph acknowledging the limitations of COI-based inference and MOTU/barcoding-gap approaches.

Minor Issues

All minor issues have been carefully addressed. Typographical errors and italicization have been corrected throughout the manuscript, figure captions have been revised to include clear labels and scale bars, and the Discussion has been reorganized to reduce repetition and to better emphasize the main message of the study.

Specific comments

L25–27 – Suggested wording change

Response:

The sentence has been revised as recommended reflecting a more cautious taxonomic interpretation, and has been incorporated into the revised manuscript.

L34 – “robust marker” wording

Response:

To prevent possible misinterpretation, the phrase “robust marker” has been replaced and incorporated according to the reviewers suggestions.

L40–54 – Introduction too long

Response:

The Introduction has been thoroughly revised by removing excessive historical background and focusing on context relevant to the present study according to the reviewer’s comment.

L80–88 – Suggested summarizing sentence

Response:

In accordance with the reviewer’s recommendation, we have added a final summarizing sentence outlining the integrative framework of the study.

L125 – Variable annealing temperatures

Response:

Yes, all the samples followed the same protocol. However, certain reasons such as DNA template sequence variation (polymorphisms) and template secondary structures can affect the annealing temperatures. Even within the same species, individuals or strains can have: SNPs (single-nucleotide polymorphisms), insertions/deletions, repetitive-sequence variability and if these variations occur near or within the primer binding sites, they: change primer–template complementarity, modify primer binding stability, make optimal annealing temperature shift up or down. Even 1 mismatch can lower the effective Tm of primer binding by 5–10°C. Similarly template secondary structure like GC-rich regions, hairpins, or repeats can also: temporarily block primer access, affect how tightly the primer hybridizes, require different temperatures for efficient binding, templates with more stable secondary structures often require higher annealing temperatures.

L137 – Bayesian Information Criterion

Response:

Bayesian Inference Criterion has been corrected to Bayesian Information Criterion.

L106 – Prioritized structures

Response:

The structures for comparisons were prioritized and mentioned accordingly as suggested by the reviewer.

L557 (Discussion) – Reduce repetition

Response:

The first introductory paragraph of discussion has been shortened and straightly focuses the key implications of our results. Similarly, the statements overstating the contents already stated in introduction and results have been removed and revised accordingly

L714 – Synonymy caution

Response:

Added a cautionary sentence noting that suggestions of synonymy are preliminary and require integrative revision.

L747 – Barcoding gap variability

Response:

We thank the reviewer for this valuable suggestion. We have now added a sentence in the Discussion acknowledging that barcoding gaps are not universal across taxa and that their detection and magnitude may vary depending on sampling depth and geographic coverage. Response to Reviewer 2

We thank Reviewer 2 for the detailed, technically insightful comments, which significantly improved the framing, reproducibility, and methodological rigor of the manuscript.

Major comments

• Title scope mismatch

Response:

We agree and have revised the title to accurately reflect the geographic and analytical scope of the study, emphasizing the Western Ghats/India focus and integrative approach.

• Introduction lacks India-focused synthesis and public health relevance

Response:

The Introduction has been restructured in a way that it:

Summarizes current knowledge of Haemaphysalis diversity in India, particularly the Western Ghats.

Explicitly highlight medical and veterinary relevance (including KFD context).

Clarify how improved taxonomy informs surveillance and disease ecology.

• Overstated taxonomic claims

Response:

All such statements in the Abstract and Discussion have been toned down and reframed as hypotheses. Taxonomic implications are now explicitly conditional on future multilocus and broader morphological evidence.

Materials and Methods

• GenBank sequence curation and quality control

Response:

We added a detailed subsection describing:

Inclusion/exclusion criteria for public sequences (length thresholds, metadata requirements, vouchers when available). Quality-control steps (translation to screen for stop codons/NUMTs, reciprocal BLAST, removal of duplicates and topological outliers). Cross-checking names against recent taxonomic revisions.

• Alignment length discrepancies and site counts

Response:

We clarified trimming rules and specified which alignment underlies each analysis. Site statistics for H. bispinosa were rechecked and corrected to ensure internal consistency, following standard definitions (S = singleton + parsimony-informative sites).

• Discussion and Conclusions

COI-only limitations and subgeneric inference

Response:

A concise cautionary paragraph was added explicitly stating that:

Inference of subgeneric non-monophyly is based on a single mitochondrial locus.

Results should be treated as preliminary hypotheses.

Potential pitfalls (misidentification, introgression, endosymbionts, NUMTs, sampling bias) are acknowledged.

Future corroboration requires nuclear markers (ITS2, 28S, 18S, H3, EF-1α) and genome-scale data.

Minor comments

Abstract/figure country counts reconciled and corrected.

DnaSP version number added.

PCR protocol verified and confirmed that Sigma-Aldrich ReadyMix™ Taq DNA Polymerase was used. The denaturation temperature has been corrected from 98 °C to 94 °C, and the PCR thermal profile has been revised accordingly in the Methods section.

UFBoot2 citation corrected to Hoang et al. (2018).

“Bayesian Inference Criterion” corrected to **Bayesian Information Criterion**.

Codon partitioning implementation in IQ-TREE and MrBayes explicitly described.

We believe that these extensive revisions have substantially strengthened the manuscript and addressed all concerns raised by both reviewers. We are grateful for the constructive feedback, which has significantly improved the clarity, rigor, and balance of our study. We hope that the revised manuscript is now suitable for publication in PLOS ONE, and we look forward to your further evaluation.

Sincerely,

K.R. Reshma

---

## [Decision Letter · Decision Letter 1]

3 Mar 2026

PONE-D-25-52984R1Integrative taxonomy of Haemaphysalis (Acari: Ixodidae) from the Western Ghats, India: COI+ morphology and implications for subgeneric boundariesPLOS One

Dear Dr. Reshma,

Thank you for submitting your manuscript to PLOS ONE. After careful consideration, we feel that it has merit but does not fully meet PLOS ONE’s publication criteria as it currently stands. Therefore, we invite you to submit a revised version of the manuscript that addresses the points raised during the review process.

We look forward to receiving your revised manuscript.

Kind regards,

Maria Stefania Latrofa

Academic Editor

PLOS One

Journal Requirements:

Additional Editor Comments:

The article has been considerably improved according to the reviewers’ suggestions; however, there are still some points that should be further revised before it can be accepted for publication.

I would suggest modifying the title again as follows: “…morphological and molecular characterization and implications.”

Line 24: change to: “were confirmed by COI sequence analyses.”

Line 37: change to: “COI for Haemaphysalis identification at the species level.”

Line 128: change to: “Molecular procedures.”

Line 209: I would suggest deleting the subheading and merging this section with the previous one into a single paragraph.

Line 235: The authority of each species should be indicated when mentioned for the first time.

Line 240: It is not clear whether “the most abundant species” refers to adults or nymphs; please specify.

Line 384: I suggest indicating the percentage of identity, even if low. Therefore, change to: “Haemaphysalis (Kaiseriana) aculeata and H. (Kaiseriana) cuspidata showed an XX% similarity with YYY, since reference COI sequences were …”

Line 387: This paragraph needs to be more concise.

Lines 439 and 446: Please indicate the values for low p-distance and high divergence.

Line 455: I suggest including data described in this paragraph in a table and summarizing it.

Reviewers' comments:

Reviewer's Responses to Questions

**Comments to the Author**

1. If the authors have adequately addressed your comments raised in a previous round of review and you feel that this manuscript is now acceptable for publication, you may indicate that here to bypass the “Comments to the Author” section, enter your conflict of interest statement in the “Confidential to Editor” section, and submit your "Accept" recommendation.

Reviewer #2: All comments have been addressed

2. Is the manuscript technically sound, and do the data support the conclusions?

Reviewer #2: Yes

3. Has the statistical analysis been performed appropriately and rigorously? 

Reviewer #2: Yes

4. Have the authors made all data underlying the findings in their manuscript fully available?

Reviewer #2: Yes

5. Is the manuscript presented in an intelligible fashion and written in standard English?

Reviewer #2: Yes

6. Review Comments to the Author

Reviewer #2: All comments and suggestions have been carefully addressed, and the manuscript has improved substantially in clarity, quality, and overall presentation.

7. PLOS authors have the option to publish the peer review history of their article (what does this mean?). If published, this will include your full peer review and any attached files.

Reviewer #2: No

---

## [Author Response · Author response to Decision Letter 2]

7 Apr 2026

Point-by-Point Response to Editor’s Comments

We sincerely thank the Editor for the constructive suggestions that have helped improve the clarity and quality of our manuscript. Below we provide our point-by-point responses.

• I would suggest modifying the title again as follows: “…morphological and molecular characterization and implications.”

Response:

Thank you for the suggestion. The title has been revised accordingly to include the phrase “morphological and molecular characterization and implications.”

• Line 24: change to: “were confirmed by COI sequence analyses.”

Response:

The sentence has been revised as suggested.

• Line 37: change to: “COI for Haemaphysalis identification at the species level.”

Response:

The sentence has been modified accordingly.

• Line 128: change to: “Molecular procedures.”

Response:

The subheading has been revised from the previous wording to “Molecular procedures,” as suggested.

• Line 209: I would suggest deleting the subheading and merging this section with the previous one into a single paragraph.

Response:

As recommended, the subheading has been removed and the text has been merged with the preceding section to form a single coherent paragraph.

• Line 235: The authority of each species should be indicated when mentioned for the first time.

Response:

Thank you for pointing this out. The taxonomic authorities for each species have now been included at their first mention in the manuscript.

• Line 240: It is not clear whether “the most abundant species” refers to adults or nymphs; please specify.

Response:

The sentence has been revised to explicitly indicate whether the statement refers to adults or nymphs.

• Line 384: I suggest indicating the percentage of identity, even if low. Therefore, change to: “Haemaphysalis (Kaiseriana) aculeata and H. (Kaiseriana) cuspidata showed an XX% similarity with YYY, since reference COI sequences were …”

Response:

Thank you for the suggestion. The sentence has been revised to include the percentage identity values. It now specifies the sequence similarity observed between Haemaphysalis (Kaiseriana) aculeata and H. (Kaiseriana) cuspidata and the closest available reference sequences.

• Line 387: This paragraph needs to be more concise.

Response: The paragraph has been edited for conciseness, removing redundant statements and improving clarity while retaining the essential information.

• Lines 439 and 446: Please indicate the values for low p-distance and high divergence.

Response:

The manuscript has been revised to explicitly include the numerical values corresponding to low p-distance and high divergence in the relevant sentences.

• Line 455: I suggest including data described in this paragraph in a table and summarizing it.

Response:

We appreciate this helpful suggestion. The data previously described in the paragraph have now been organized into a new table to improve clarity and readability. The text has been shortened accordingly to provide a brief summary referring to the table.

---

## [Editor Report · Decision Letter 2]

19 Apr 2026

Integrative taxonomy of Haemaphysalis (Acari: Ixodidae) from the Western Ghats, India: morphological and molecular characterization and implications

PONE-D-25-52984R2

Dear Dr. Reshma,

We’re pleased to inform you that your manuscript has been judged scientifically suitable for publication and will be formally accepted for publication once it meets all outstanding technical requirements.

Kind regards,

Maria Stefania Latrofa

Academic Editor

PLOS One

---

## [Editor Report · Acceptance letter]

PONE-D-25-52984R2

PLOS One

Dear Dr. Reshma,

I'm pleased to inform you that your manuscript has been deemed suitable for publication in PLOS One. Congratulations! Your manuscript is now being handed over to our production team.

Kind regards,

on behalf of

Dr. Maria Stefania Latrofa

Academic Editor

PLOS One